# Relationship between the morphological, mechanical and permeability properties of porous bone scaffolds and the underlying microstructure

Yongtao Lu[1,2,3]*, LiangLiang Cheng[4]*, Zhuoyue Yang[1], Junyan Li[5], Hanxing Zhu[6]

1 Department of Engineering Mechanics, Dalian University of Technology, Dalian, China, 2 State Key Laboratory of Structural Analysis for Industrial Equipment, Dalian University of Technology, Dalian, China, 3 DUT-BSU Joint Institute, Dalian University of Technology, Dalian, China, 4 Affiliated Zhongshan Hospital of Dalian University, Dalian, Liaoning, China, 5 School of Science and Technology, Middlesex University, London, United Kingdom, 6 School of Engineering, Cardiff University, Cardiff, United Kingdom

* yongtaolu@dlut.edu.cn (YL); liangliang30766@163.com (LLC)

**Data Availability Statement:** The datasets used in the present study are available at https://doi.org/10.6084/m9.figshare.12721325.

## Abstract

Bone scaffolds are widely used as one of the main bone substitute materials. However, many bone scaffold microstructure topologies exist and it is still unclear which topology to use when designing scaffold for a specific application. The aim of the present study was to reveal the mechanism of the microstructure-driven performance of bone scaffold and thus to provide guideline on scaffold design. Finite element (FE) models of five TPMS (Diamond, Gyroid, Schwarz P, Fischer-Koch S and F-RD) and three traditional (Cube, FD-Cube and Octa) scaffolds were generated. The effective compressive and shear moduli of scaffolds were calculated from the mechanical analysis using the FE unit cell models with the periodic boundary condition. The scaffold permeability was calculated from the computational fluid dynamics (CFD) analysis using the 4×4×4 FE models. It is revealed that the surface-to-volume ratio of the Fischer-Koch S-based scaffold is the highest among the scaffolds investigated. The mechanical analysis revealed that the bending deformation dominated structures (e.g., the Diamond, the Gyroid, the Schwarz P) have higher effective shear moduli. The stretching deformation dominated structures (e.g., the Schwarz P, the Cube) have higher effective compressive moduli. For all the scaffolds, when the same amount of change in scaffold porosity is made, the corresponding change in the scaffold relative shear modulus is larger than that in the relative compressive modulus. The CFD analysis revealed that the structures with the simple and straight pores (e.g., Cube) have higher permeability than the structures with the complex pores (e.g., Fischer-Koch S). The main contribution of the present study is that the relationship between scaffold properties and the underlying microstructure is systematically investigated and thus some guidelines on the design of bone scaffolds are provided, for example, in the scenario where a high surface-to-volume ratio is required, it is suggested to use the Fischer-Koch S based scaffold.

**Funding:** Dr. Yongtao Lu received the funding from the National Natural Science Foundation of China (grant number: 11702057), the Liaoning Provincial Natural Science Foundation of China (grant number: 2019-MS-040) and the DUT-BSU grant (grant number: ICR1903). Dr. Yongtao Lu and Dr. Hanxing Zhu jointly received the funding from the State Key Laboratory of Structural Analysis for Industrial Equipment, Dalian University of Technology (GZ19108). These funding sponsored Dr. Lu and Dr. Zhu's research activities including study design, data analysis and preparation of the manuscript.

**Competing interests:** The authors declare that they have no conflict of interest.

## Introduction

In recent years, due to the increased human life expectancy and the increased number of bone diseases and traumas [1], there has been an increasing demand for organ transplantations and consequently a high demand for new artificial tissue substitutes [2]. Porous scaffolds are considered to be one of the best candidates for bone substitute materials because macroscopically the scaffold stiffness can be tuned to match to that of the human bones and microscopically the porous structure can facilitate the cell behaviors [3–5]. Additionally, the emerging novel manufacturing technologies, such as the additive manufacturing, enable the productions of porous scaffolds with complex micro-architectures [6–8]. However, designing optimized tissue scaffolds is still a challenging work due to the conflict in the mechanical and biological needs of scaffolds [9]. For instance, high porosity is a desirable property in satisfying the biological requirements, but such attribute reduces the mechanical compatibility of scaffolds, such as the effective modulus, the failure strength and the fatigue life [10, 11].

Recent studies have showed that not only the pore size and porosity, but also the curvature of pores, pore shape, etc. play an important role in the performance of porous scaffolds [12–15], which makes the design of scaffold microstructure a crucial step. Because the scaffold properties are mainly determined by the scaffold topology (i.e., the type of unit cell), the selection of an appropriate scaffold topology becomes an essential step in the relevant fields. With regard to the scaffold topology, recent years have seen the design trend moving from the traditional type (cube, Octa, Octet, etc.) to the Triple Periodic Minimal Surface (TPMS)-based type (Diamond, Gyroid, etc.) [16]. One of the reasons is that the traditional scaffolds have sharp convex edges and corners, which is not preferred in the tissue growth process [17]. On the contrary, the TPMS-based scaffolds have a mean curvature of zero [18], a high surface-to-volume ratio [19], the ease of functional grading [20] and a variable /tunable electrical/thermal conductivity [21], which can make their properties anatomical location-specific and subject-specific and consequently can largely increase their potentials in the applications in biomedicine and relevant fields [22]. In recent years, the functionally graded scaffold and other novel design strategy has been used to design scaffolds [23–28], because the bionic scaffolds, which have the mechanical and biological properties similar to those of the replaced natural tissues, can be achieved using these methods. However, when designing uniform or functionally graded scaffolds for a specific application, it is still unclear which scaffold topology is the best candidate among the vast TPMS scaffold topologies, e.g., Gyroid, Diamond, Schwarz P. This is because the relationship between the scaffold properties and the underlying topologies is still unclear.

To understand the relationships between the scaffold properties and the underlying topologies, a number of experimental and numerical studies have been performed in recent years [3, 11, 29–36]. For examples, Egan et al. (2017) compared the mechanical and permeability behaviors of eight traditional scaffolds (Cube, BC-Cube, Octet, etc.); Zhao et al., (2018) evaluated the effect of tetrahedron and octahedron pore geometries on the fatigue and cell affinity behaviors of porous scaffolds; Wang et al., (2018) investigated the effect of various diamond crystal lattices on osteointegration and osteogenesis; Almeida and Bartolo (2014) evaluated the mechanical behaviors of two TPMS-based scaffolds, namely the Schwarz and Schoen and Maskery et al. (2018a) compared the mechanical behaviors of three TPMS-based scaffolds, i.e., Gyroid, Diamond and Primitive. These studies provided valuable data for the scaffold design. However, these studies only investigated either the traditional scaffolds [30] or just several TPMS-based scaffolds [29], or only focused on one property of the scaffolds [31, 32]. A comprehensive comparison of the morphological, mechanical and permeability properties of different TPMS-based scaffolds is still missing and the mechanism of the microstructure-driven scaffold

properties is still unclear. This comprehensive comparison will help the selection of appropriate scaffold topologies in the design of uniform or functionally graded scaffolds.

The aims of the present study were to reveal the mechanism of the microstructure-driven performance of bone scaffolds and thus to provide the guideline on scaffold design.

## Materials and methods

### The finite element models of bone scaffolds

Finite element (FE) models of five widely-used TPMS-based scaffolds and three traditional scaffolds were generated. When evaluating the mechanical properties of the scaffolds, the scaffold unit cell model with the periodic boundary condition was used. The unit cell (one representative periodic microstructure) models of the TPMS-based scaffolds were generated following the methodology presented in the literature [37, 38]. In brief, the software of K3DSurf developed by Abderrahman Taha (http://k3dsurf.sourceforge.net) was used to generate the surface models of the unit cell with the dimension of $2.5 \times 2.5 \times 2.5$ mm$^3$ (Fig 1A). Afterwards, the unit cell surface models were imported into SolidWorks 2017 (Dassault Systemes SolidWorks Corporation, Waltham, MA) to generate the TPMS network solid model, where the domain to one side of the TPMS represents the solid material and the other side

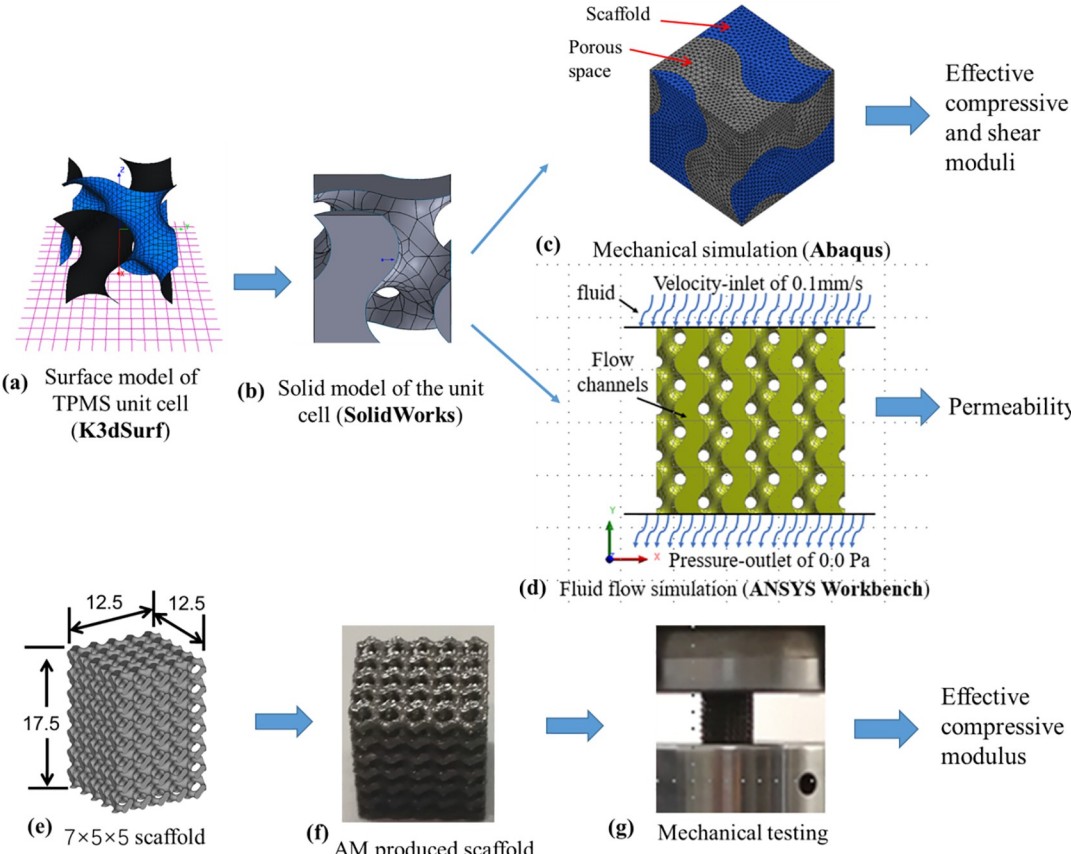

**Fig 1.** Workflow for analyzing the mechanical properties and permeability of TPMS-based scaffolds: (a) and (b) generation of the solid model of the TPMS unit cell, (c) calculation of the effective compressive and shear moduli of the scaffolds using unit cell model with periodic boundary conditions, (d) calculation of the scaffold permeability using the computational fluid dynamics analysis, (e) and (f) design and additive manufactured scaffold, and (g) mechanical testing of the scaffold.

**Fig 2. Unit cell models of five TPMS-based (left) and three traditional (right) scaffolds analyzed in the present study.**

represents the void domain (Fig 1B). The geometric models were then imported into ABAQUS (Version 6.13, Dassault Systems SIMULIA Ltd, Providence, RI), where the FE meshes were generated and the FE calculations were performed (Fig 1C). To facilitate the application of periodic boundary condition, the void domain within the unit cell model was also discretized into FE meshes. The unit cell model, including the solid and the void domains, was meshed using the second-order three-dimensional (3D) tetrahedral elements (C3D10) (Fig 1C). The unit cell models of the traditional scaffolds with the same dimension, i.e., $2.5 \times 2.5 \times 2.5$ mm$^3$, were generated in ABAQUS by giving the dimensions of the struts (i.e., height, width, and diameter).

When evaluating the permeability of the scaffolds, the unit cell model was assembled to form the 4×4×4 FE models in ANSYS Workbench (Release 15.0.3, ANSYS Inc., Cannonsburg, PA) (Fig 1D). The number of repetition of the unit cell was selected based on the criterion that the computational time should be efficient and the errors induced by the boundary conditions should be minimal [39]. A sensitivity study showed that the relative difference between the permeability calculated from the 4×4×4 and the 5×5×5 FE models was as small as 0.1% (i.e., the boundary effect has been removed). Therefore, the 4×4×4 FE model was used in the present study.

In the present study, the TPMS models of Diamond, Gyroid, Schwarz P, Fischer-Koch S and F-RD were created (Fig 2), because they are the basic TPMS unit cells [12] and are of high interest in the biomedicine and relevant fields [29, 31, 32]. The approximated periodic nodal equations for the five TPMS scaffolds are presented in Table 1 and different scaffold porosities can be obtained by changing the constant (C) in the TPMS nodal equations. The traditional scaffolds of Cube, FD-Cube and Octa were created (Fig 2), because they have representative mechanical and permeability behaviors compared to other traditional scaffolds [30].

## The morphological, mechanical and permeability properties of scaffolds

The scaffold porosity ($\emptyset$) and the surface-to-volume ratio ($S/V$) were calculated to describe the morphological properties of scaffolds. Scaffold porosity was calculated as the value using the scaffold void volume divided by its nominal volume (i.e., the volume of the cube encompassing

**Table 1. The periodic nodal equation for the TPMS structures investigated in the present study.**

| TPMS structures | Nodal equations ($x, y$ and $z$ are the nodal coordinates and C is a constant) |
|---|---|
| Diamond | $U_D = \sin(x)\sin(y)\sin(z) + \sin(x)\cos(y)\cos(z) + \cos(x)\sin(y)\cos(z) + \cos(x)\cos(y)\sin(z)\text{—C}$ |
| Gyroid | $U_G = \cos(x)\sin(y)+\cos(y)\sin(z)+\cos(z)\sin(x)\text{—C}$ |
| Schwarz P | $U_S = \cos(x) + \cos(y) + \cos(z)\text{—C}$ |
| Fischer-Koch S | $U_F = \cos(2x)\sin(y)\cos(z) + \cos(2y)\sin(z)\cos(x) + \cos(2z)\sin(x)\cos(y)\text{—C}$ |
| F-RD | $U_R = 8\cos(x)\cos(y)\cos(z) + \cos(2x)\cos(2y)\cos(2z)-\cos(2x)\cos(2y) + \cos(2y)\cos(2z)+\cos(2z)\cos(2x)\text{–C}$ |

the scaffold) and $S/V$ was calculated as the value using the scaffold inner surface area divided by the nominal volume.

The normalized effective compressive ($E_c$) and shear moduli ($G_s$) were calculated to describe the mechanical behaviors of scaffolds. When evaluating the effective elastic moduli, the base material of Ti-6Al-4V was chosen for the scaffold. Therefore, a Young's modulus of 110.0 GPa [40] and a Poisson's ratio of 0.34 were defined for the solid domain and no nonlinear mechanical properties were defined because of the linear elastic simulation. In the FE unit cell model, a Young's modulus of 1.0 MPa and a Poisson's ratio of 0.45 [41] were defined for the void domain to facilitate the definition of periodic boundary condition, because in some unit cell models, the scaffold solid phase finished at one exterior surface and there were no corresponding elements in the opposite surface. Using the FE unit cell model, the effective elasticity tensor for each scaffold was first derived by solving the material constitutive equations, established by defining three individual loading, i.e., $\varepsilon_x = 0.01$, $\varepsilon_y = 0.01$, $\varepsilon_{xy} = 0.01$ [42, 43]. The reason for defining three individual loading is that all the scaffolds investigated have three nonzero constants in the elasticity matrix and can be regarded as the structure with a cubic symmetry [42]. In the loading scenarios, while one strain component was applied, other strain components were left free. The effective compressive ($E_e$) and shear moduli ($G_e$) of scaffold were then calculated from the elasticity tensors [44]. To eliminate the influence of the base material, the effective compressive ($E_e$) and shear moduli ($G_e$) were normalized to the compressive and shear moduli of Ti-6Al-4V, respectively. The normalized effective compressive ($E_c$) and shear moduli ($G_s$) were formulated as below:

$$E_c = E_e / E_{Ti} \tag{1}$$

$$G_s = G_e / G_{Ti} \tag{2}$$

where, $E_e$ and $G_e$ are the effective compressive and shear moduli of the scaffold; $E_{Ti}$ (110.0 GPa) and $G_{Ti}$ (41.05 GPa) are the elastic and shear moduli of Ti-6Al-4V. When calculating the mechanical properties, a mesh convergence study was performed to ensure that the influence of the mesh size on the FE predicted compressive and shear moduli was less than 0.5% (regarded as converged) (Fig 3) and the element size of approximately 0.1 mm was used,

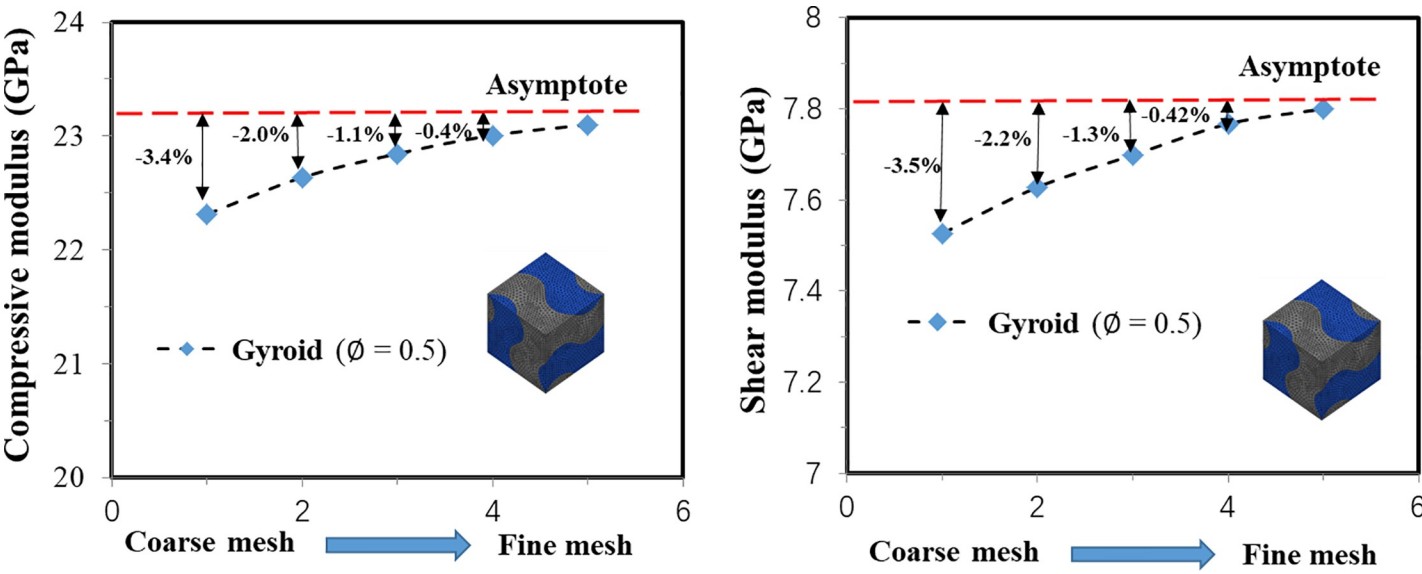

**Fig 3. Demonstration of the influence of mesh size on the scaffold mechanical property (mesh convergence) using the Gyroid with the porosity of 0.5.**

which resulted in approximately 0.06 million elements for the Fischer-Koch S unit cell model with the porosity of 0.55.

The scaffold permeability ($k$) was calculated to evaluate the biological behaviors of the scaffolds. The FLUENT module in ANSYS Workbench was used to perform the Computational Fluid Dynamics (CFD) analysis. When calculating the permeability, the scenario of unidirectional fluid flow going through the scaffold was simulated and the FE model of the void spaces within the solid scaffold was created. The void domains were meshed using the tetrahedral element with the size of approximately 0.2 mm, which resulted in approximately 0.64 million elements for the Fischer-Koch S scaffold with the porosity of 0.55. The following boundary conditions were defined when calculating the scaffold permeability: walls were placed around the four sides of the model and in the areas which were in contact with the solid scaffold, to represent a flow channel (Fig 3C); the fluid with a velocity of 0.1 mm/s was assigned to the inlet of the scaffold; zero gauge pressure was adopted on the outlet, and no-slip conditions was imposed [45]. The fluid was modeled as the incompressible water with a viscosity of 0.001 Pa and a density of 998.2 kg/m$^3$ [45] and the scenario of laminar flow was simulated. The scaffold permeability ($k$) was determined using the Darcy's relationship [45, 46] as follows:

$$k = \frac{Q\mu L}{A\Delta P} \tag{3}$$

where, $Q$ is the fluid flow rate, $\mu$ is the dynamic fluid viscosity, $L$ is the length of the cubic scaffold, $A$ is the cross-sectional area of the scaffold and $\Delta P$ is the pressure drop (units of Pa). The pressure drop was calculated by using the average pressure at the inlet (the pressure at the outlet was zero) and then the scaffold permeability was calculated using Eq (3).

Additionally, for comparison, the scaffold permeability was calculated based on the Kozeny-Carmen's empirical formula [47]:

$$k^* = \frac{\emptyset_1^3}{2s^2} \tag{4}$$

where, $\emptyset_1$ is the scaffold porosity and $s$ is the ratio of the inner pore surface area to the total volume of the sample.

The relationships between the mechanical properties of the scaffold and the scaffold porosity and between the permeability and the scaffold porosity were investigated. Regarding the mechanical property, to reflect the underlying physical phenomena, the relationship between the relative elastic compressive modulus and the scaffold volume fraction were described using the exponential function proposed by Gibson and Ashby [31, 32]:

$$E^* = C_1\rho^n + E_0 \tag{5}$$

where, $\rho$ is the relative volume fraction of the scaffold ($\rho = 1-\emptyset$), $E_0$ is the offset of the elastic modulus, $C_1$ and $n$ are the material constants. The values of $C_1$, $n$ and $E_0$ were obtained by fitting Eq (5) to the relationship curve of the scaffold elastic modulus and the volume fraction. It is reported in the literature that the value of the prefactor $C_1$ is in the range from 0.1 to 4.0, the value of $n$ is approximately 2.0 when the deformation of the cellular struts is bending-dominated and $n$ is approximately 1.0 when the deformation is stretching-dominated [31, 32]. In this paper, the values of $C_1$, $n$ and $E_0$ were determined and then the deformation features for different scaffolds were discussed. Regarding the scaffold permeability and the surface-to-volume ratio, the statistical regression equations (quadratic or other forms) and the coefficient of determinations ($R^2$) were computed for the relationships between them and the scaffold porosity. The reasons for deriving these statistical regression equations are to enable the

interpolation of the data points to the full scaffold porosity range and to facilitate the scaffold design by using these relations. In the present study, for each type of scaffold, five FE models with different porosities were created, the corresponding mechanical and permeability properties were obtained and then the values at other porosities were worked out by the interpolation using the derived fitting equations.

### Validation of the predictions of the FE models of scaffolds

The compressive elastic modulus of the scaffold predicted from the FE simulation was validated using the mechanical testing data. The Gyroid and Diamond-based scaffolds were selected and three scaffold porosities between 50% and 80% per TPMS type were designed. The dimension of the unit cell model was 2.5×2.5×2.5 mm$^3$ and the dimension of the scaffold sample was 17.5×12.5×12.5 mm$^3$ (7×5×5 unit cells) (Fig 1E). The designed Gyroid and Diamond-based scaffolds were produced using the additive manufacturing method of Selective Laser Melting (SLM) (Renishaw AM250, Renishaw plc., Gloucestershire, UK) with the scanning speed of 0.04 m/s, the laser power of 350.0 W and the hatch angle of 90 degrees (Fig 1F). The defected scaffolds, i.e. the deviation of the porosity from the design value is larger than 5%, were disposed and five samples per scaffold porosity were selected. Then the scaffolds were placed on the MTS Landmark® Servohydraulic Test Systems (MTS Systems Corporation, Eden Prairie, MN) and the quasi-static testing was performed, where the crosshead speed was 0.5 mm/min (Fig 1G). The effective compressive moduli were calculated from the mechanical testing and used to validate the predictions from the FE analysis.

## Results

### Validation of the FE models of scaffolds

A representative stress-strain curve from the mechanical testing of scaffold is presented in Fig 4. The effective compressive moduli of the Gyroid and Diamond-based scaffolds predicted from the FE analysis were compared to those obtained from the experimental testing (Table 2). For both the Gyroid and Diamond based scaffolds, the differences between the FE and experimental values (using the experimental data as the reference) are within 10% (Table 2).

### The effective compressive and shear properties of scaffolds

The relationships between the normalized effective compressive modulus and the porosity, between the normalized effective shear modulus and the porosity are plotted in Fig 5. The compressive moduli of the Schwarz P and Cube-based scaffolds are the highest, followed by the FD-Cube, the Octa, the Fischer-Koch S, the Gyroid, the F-RD and the Diamond-based scaffolds. The shear moduli of the Diamond-based scaffold are the highest, followed by the F-RD, the Gyroid, the Fischer-Koch S, the Octa, the FD-Cube, the Schwarz P and the Cube-based scaffolds (Fig 5B).

For each type of scaffold, the normalized shear modulus is highly linearly correlated with the normalized compressive modulus ($R^2 > 0.99$) (Fig 5C). The slopes of all the linear regression lines are larger than one. The slope for the Diamond-based scaffold is the highest ($G_s = 3.16 E_c + 0.03$), followed by the F-RD ($G_s = 2.73 E_c + 0.02$), the Gyroid ($G_s = 2.25 E_c + 0.03$), the Fischer-Koch S ($G_s = 2.51 E_c + 0.03$), the Octa ($G_s = 1.65 E_c + 0.07$), the FD-Cube ($G_s = 2.73 E_c + 0.02$), the Schwarz P ($G_s = 1.59 E_c$—0.09) and the Cube ($G_s = 1.64 E_c$—0.17) based scaffolds. On the other hand, except for the cube-based scaffold, almost all the points investigated in the present study lies in the upper side of the diagonal line (i.e., $G_s = E_c$).

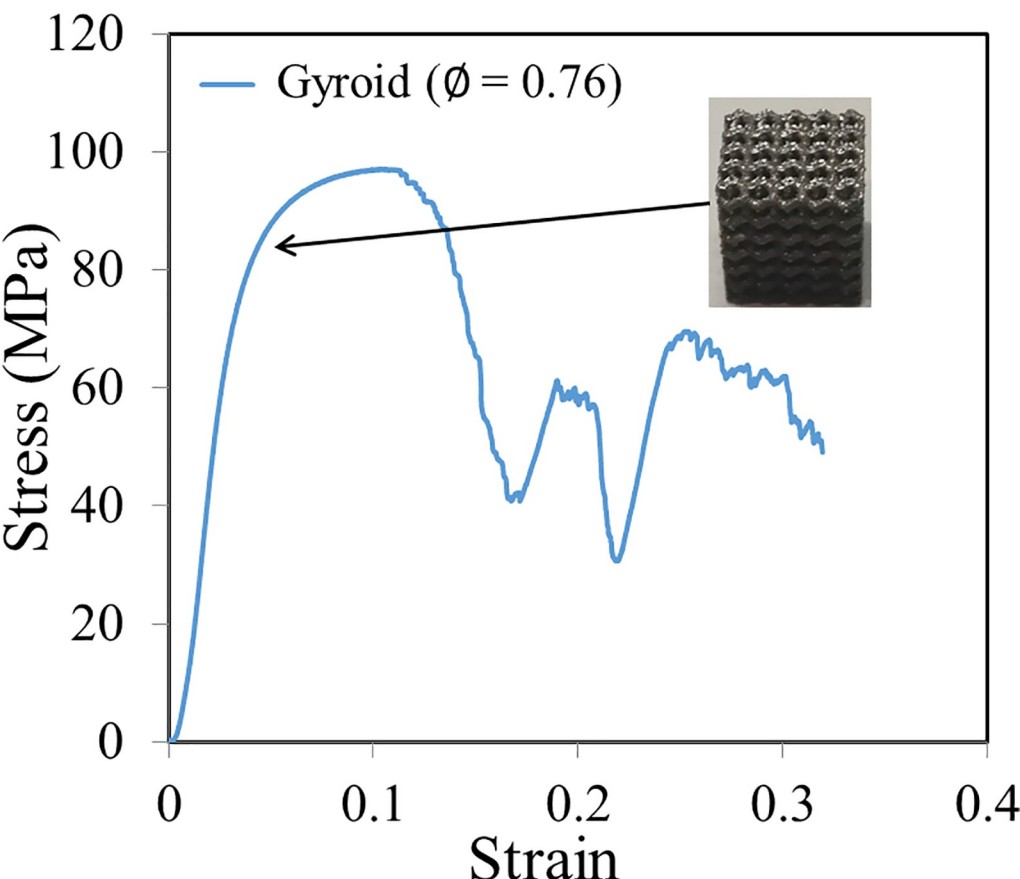

**Fig 4. A representative stress-strain curve obtained from the mechanical testing of additively manufactured scaffold.**

The values of $C_1$, $n$ and $E_0$ for different scaffolds are presented in Table 3. The values of $C_1$, $n$ and $E_0$ for the Diamond, Gyroid and Fischer-Koch S based scaffolds agree well with the values reported in literature [31, 32]. The values of the parameter $C_1$ for all the scaffolds are between 0.1 and 4.0, which is also in good agreement with the literature [31, 32]. The values of $n$ for the Diamond, the Gyroid, the F-RD and the Fischer-Koch S based scaffolds are close to 2.0, while they are close to 1.0 for the Schwarz P, Cube, the FD-Cube and the Octa based scaffolds.

**Table 2. Comparison of the effective compressive moduli of the TPMS-based scaffolds predicted from the finite element analysis with the experimental testing data (presented as the mean ± standard deviation, 5 samples per porosity per topology).**

|  |  | Porosity = 0.51 | Porosity = 0.67 | Porosity = 0.76 |
|---|---|---|---|---|
| Gyroid | Experiment (n = 5) | 19.84 ± 0.81 GPa | 8.39 ± 0.72 GPa | 3.98 ± 0.62 GPa |
|  | FE prediction | 21.59 GPa | 9.09 GPa | 4.37 GPa |
|  | Difference (%) | 8.82% | 8.30% | 9.73% |
|  |  | Porosity = 0.54 | Porosity = 0.66 | Porosity = 0.79 |
| Diamond | Experiment (n = 5) | 15.78 ± 0.73 GPa | 8.07 ± 0.62 GPa | 3.17 ± 0.51 GPa |
|  | FE prediction | 16.51 GPa | 8.71 GPa | 3.42 GPa |
|  | Difference (%) | 4.63% | 7.93% | 7.89% |

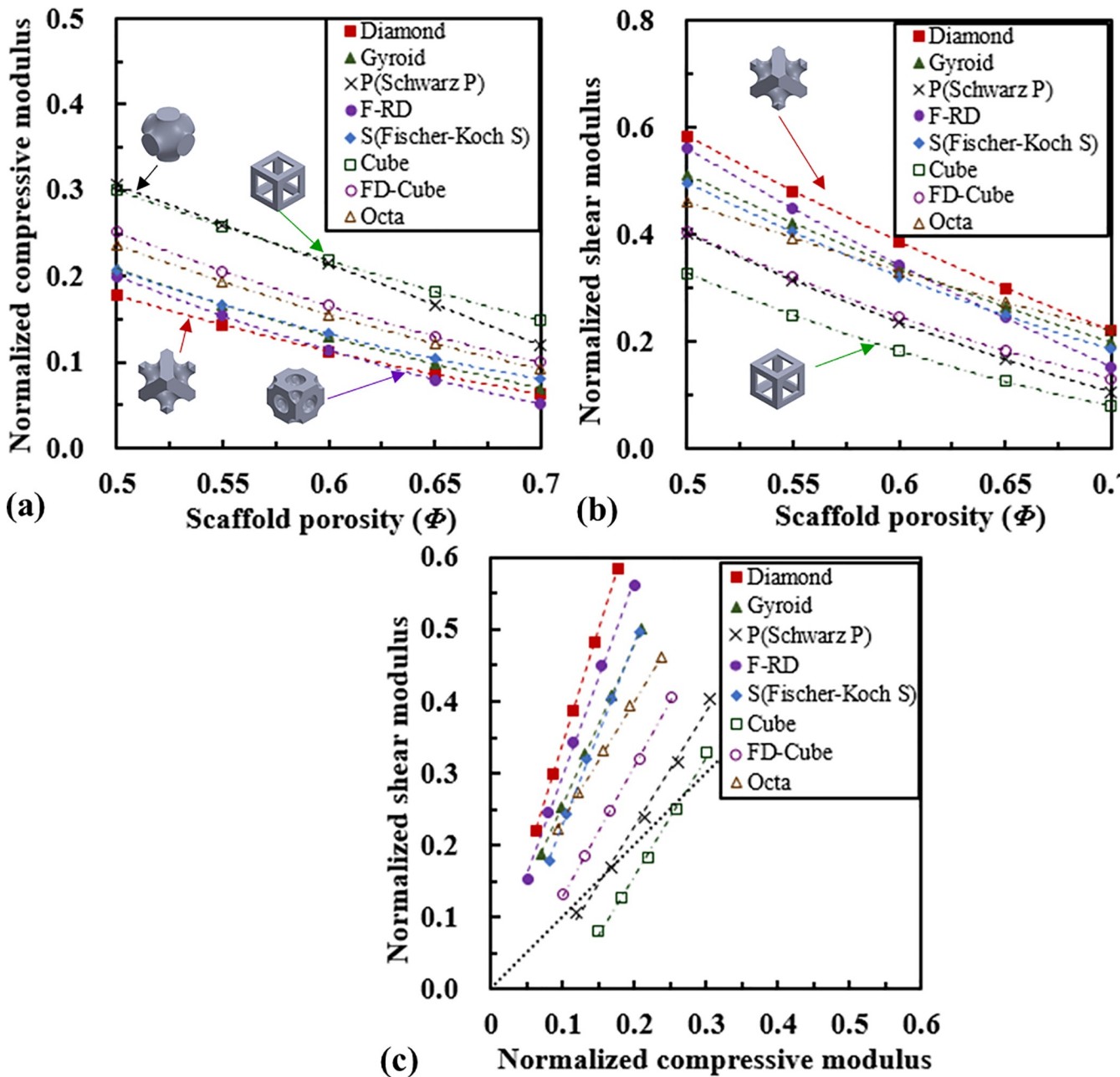

**Fig 5.** Comparison of the compressive and shear properties of scaffolds: (a) the relationship between the normalized effective compressive modulus and scaffold porosity, (b) the relationship between the normalized effective shear modulus and scaffold porosity, and (c) the relationship between the normalized effective shear and compressive moduli.

### The surface-to-volume ratio and permeability properties of scaffolds

The relationship between the surface-to-volume ratio ($S/V$) of the scaffolds and the porosity is presented in Fig 6. For all the scaffolds except the Octa-based one, the surface-to-volume ratio is not a monotonic function of the porosity and the surface-to-volume ratios are the highest when the porosity is 0.5, and start to decrease when the porosity is away from 0.5, the reason for which could be that the overlapped inner surfaces increase when the porosities get lower and there are fewer inner surfaces with the increase of the scaffold porosity. The surface-to-

**Table 3. Gibson-Ashby parameters for different scaffolds; all fittings have $R^2 > 0.99$.**

| Scaffold type | | $C_1$ | $n$ | $E_0$ | |
|---|---|---|---|---|---|
| Diamond | The present study | 0.743 | 2.081 | 0.0021 | Bending |
| | Maskery et al., 2018a | 0.750 | 2.102 | 0.0032 | |
| Gyroid | The present study | 0.919 | 2.131 | -0.0011 | Bending |
| | Maskery et al., 2018a | 1.020 | 2.405 | 0.0021 | |
| Schwarz P | The present study | 0.950 | 0.863 | -0.2193 | Stretching |
| | Maskery et al., 2018a | 0.920 | 1.001 | -0.1724 | |
| F-RD | The present study | 0.955 | 1.920 | -0.0472 | Bending |
| Fischer-Koch S | The present study | 0.918 | 2.322 | 0.0242 | Bending |
| Cube | The present study | 0.672 | 1.216 | 0.0001 | Stretching |
| FD-Cube | The present study | 0.689 | 1.361 | -0.0002 | Stretching |
| Octa | The present study | 0.523 | 1.474 | 0.0002 | Stretching |

volume ratio of the Fischer-Koch S-based scaffold is the highest, followed by the F-RD, the FD-Cube, the Diamond, the Gyroid, the Cube and the Schwarz P-based scaffolds. Quadratic relationships were found between the surface-to-volume ratio and the porosity (Table 4) and all the fits have $R^2 > 0.99$. The interpolated values using the fitted quadratic relationships and the comparison of the surface-to-volume ratio in the porosity range from 0.3 to 0.7 are presented in Fig 6B.

The relationship between the scaffold permeability (calculated from Darcy's law) and the porosity is presented in Fig 7A. The permeability of the Cube-based scaffold is the highest, followed by the Schwarz P, the Gyroid, the Diamond, the FD-Cube, the Octa, the F-RD and the Fischer-Koch S-based scaffolds. Quadratic relationships are not always the best to describe the relationships between the scaffold permeability and the scaffold porosity. To achieve a high coefficient of determination ($R^2$), different relationships have to be used for different

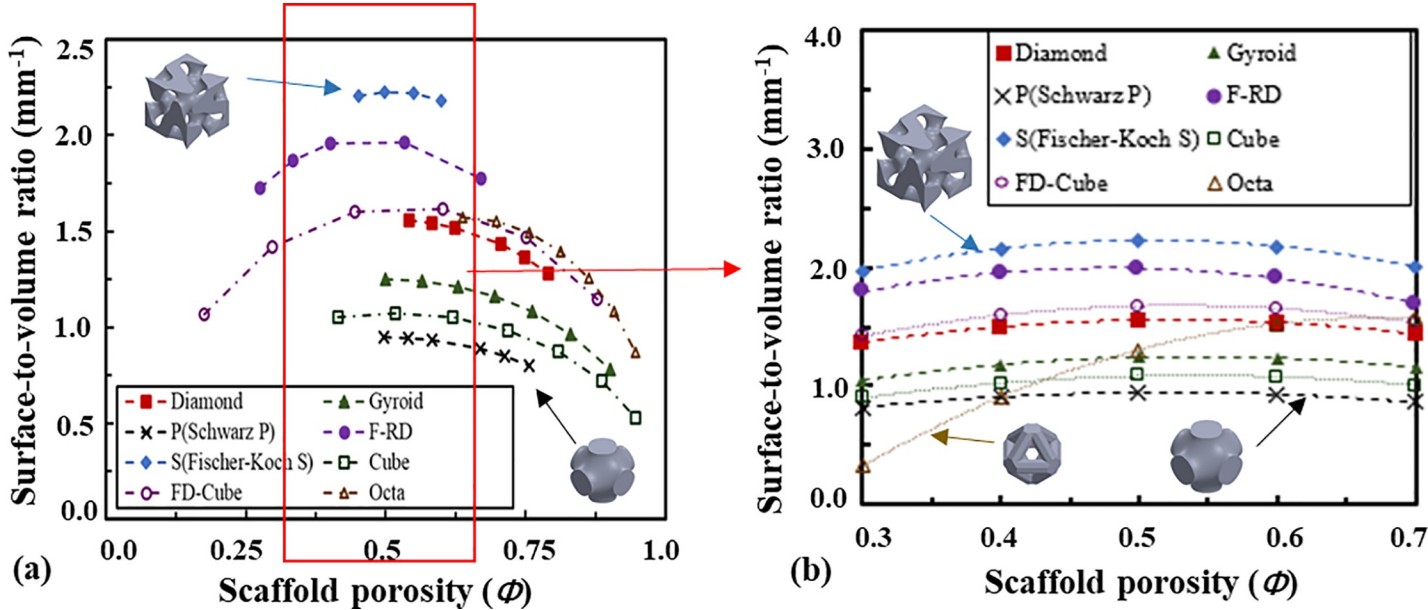

**Fig 6.** Relationship between the surface-to-volume ratio of the scaffold and the scaffold porosity: (a) comparison in the full range of the scaffold porosity and (b) the interpolated values and comparison in the range of scaffold porosity from 0.3 to 0.7.

**Table 4. The regression equations for the scaffold surface-to-volume ratio and the permeability as a function of the scaffold porosity (denoted as $\emptyset$ and ranged from 0.0 to 1.0).**

|  | Surface-to-volume ratio [mm$^{-1}$] | Permeability [$10^{-7}$ m$^2$] |
|---|---|---|
| Diamond | $S/V = -3.87\emptyset^2 + 4.05\emptyset + 0.49$ (R$^2$ = 0.996) | $k = 0.01\,e^{5.05\emptyset}$ (R$^2$ = 0.992) |
| Gyroid | $S/V = -3.52\emptyset^2 + 3.80\emptyset + 0.22$ (R$^2$ = 0.994) | $k = 0.01\,e^{4.87\emptyset}$ (R$^2$ = 0.991) |
| Schwarz P | $S/V = -2.62\emptyset^2 + 2.73\emptyset + 0.24$ (R$^2$ = 0.995) | $k = 0.01\,e^{5.55\emptyset}$ (R$^2$ = 0.992) |
| F-RD | $S/V = -6.16\emptyset^2 + 5.91\emptyset + 0.57$ (R$^2$ = 0.994) | $k = 0.89\emptyset^2 - 0.48\emptyset + 0.07$ (R$^2$ = 0.995) |
| Fischer-Koch S | $S/V = -5.98\emptyset^2 + 6.08\emptyset + 0.68$ (R$^2$ = 0.994) | $k = 0.59\emptyset^2 - 0.29\emptyset + 0.04$ (R$^2$ = 0.996) |
| Cube | $S/V = -3.26\emptyset^2 + 3.53\emptyset + 0.14$ (R$^2$ = 0.995) | $k = 0.01\,e^{5.47\emptyset}$ (R$^2$ = 0.991) |
| FD-Cube | $S/V = -4.33\emptyset^2 + 4.67\emptyset + 0.39$ (R$^2$ = 0.995) | $k = 1.25\emptyset^{4.20}$ (R$^2$ = 0.992) |
| Octa | $S/V = -9.02\emptyset^2 + 12.13\emptyset - 2.50$ (R$^2$ = 0.995) | $k = 0.01\,e^{5.27\emptyset}$ (R$^2$ = 0.991) |

topologies as presented in Table 4 and all the fits have R$^2$ > 0.99. Regarding whether the scaffold permeability correlates with its deformation mechanism (i.e., whether the stretching deformation dominated structures have a higher permeability than the bending deformation dominated structures, or vice versa), no trend is found in the present study.

The relationship between the permeability calculated from the Kozeny-Carman's relation and that calculated from the Darcy's law is presented in Fig 7B. All the points lies in the upper side of the diagonal line ($k^* = k$). When the scaffold permeability increased (i.e., porosity increased), the points moved further away from the diagonal line. Corresponding to the permeability calculated from Darcy's law, the one calculated from Kozeny-Carman's relation is the highest for the Schwarz P-based scaffold, followed by the Gyroid, Diamond, the Fisher-Koch S, the Octa, the FD-Cube and the Cube based scaffolds (Fig 7B).

## Discussion

In the present study, the morphological, mechanical and permeability properties of five commonly used TPMS-based scaffolds and three traditional scaffolds were analyzed using the finite element analysis for the aim to provide some guidelines on the design of bone scaffold.

The validation of the FE predictions is necessary, because the scaffolds are produced using the additive manufacturing technique (SLM), in which process the properties of the scaffolds can be influenced by the build orientation, the un-melted powders, the discrepancy between the produced scaffold geometry and the nominal Computer-aided Design (CAD) input, etc. [48, 49]. Therefore, the validation is to assure that the FE models developed in the present study can well predict the mechanical behavior of additively manufactured scaffolds. Nevertheless, an acceptable discrepancy of within 10.0% is found between the FE predictions and the experimental results, which is in the same order as that reported in the literature [48]. In the present study, the experimentally measured compressive elastic moduli are smaller than the FE predicted results, which could be caused by the partially melted and imperfectly bonded powders in the produced scaffolds. The permeability predicted from the CFD analysis performed in the present study is not validated. However, previous studies have found a highly linear correlation between the permeability derived from CFD analysis and the experimental results (with a factor of approximately 0.27), concluding that the CFD analysis is a reliable tool for estimating the scaffold permeability [50, 51]. This is confirmed by the fact that the range of permeability ($1.0 \times 10^{-10}$ m$^2$ to $1.0 \times 10^{-8}$ m$^2$) predicted in the present study agrees with the experimental data using the flow chambers [52, 53].

The analysis on the Gibson-Ashby fitting revealed that the deformations of the Diamond, the Gyroid, the F-RD and the Fischer-Koch S based scaffolds are bending dominated, while

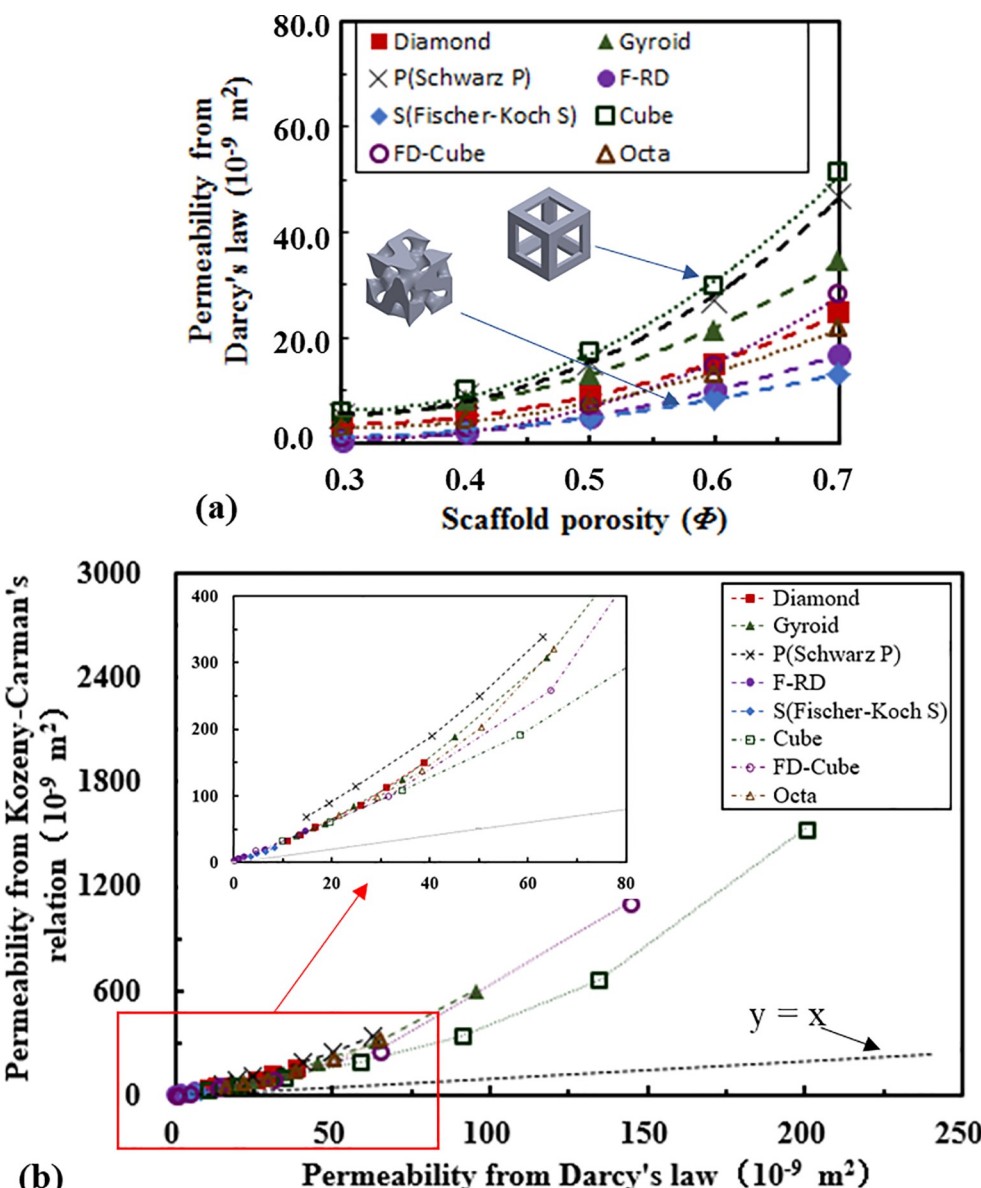

**Fig 7.** Comparison of the permeability of scaffolds: (a) the relationship between the scaffold permeability (calculated from Darcy's law) and porosity, and (b) the relationship between the permeability calculated from Kozeny-Carman's relation and that from Darcy's law.

the deformations of the Schwarz P, the Cube, the FD-Cube and the Octa based scaffolds are stretching dominated. This mechanism can be explained by the arrangement of the scaffold microstructure, for example, in the Schwarz P based scaffold, the main beams/structures are aligned in the compressive/tensile loading direction and consequently its ability to resist compression and tension is high. The bending or compression mechanism is also reflected in the scaffold modulus-porosity relationship curves (Fig 5), i.e., at the same porosity, the stretching dominated structures (e.g., Cube, Schwarz P) have higher effective compression moduli, while the bending dominated structures (e.g., Diamond) have higher effective shear moduli. The values of $C_1$, $n$ and $E_0$ obtained for the Diamond, the Gyroid and the Schwarz P based scaffolds

agree well with the literature data [31, 32], reflecting the appropriate settings in the FE analysis performed in the present study.

Regarding the mechanical and permeability properties of different TPMS-based scaffolds, it is revealed that different scaffolds exhibit different properties, making them suitable for different applications. For example, the Schwarz P-based scaffold has the straight and smoothed structures aligned in the compressive loading direction, and consequently the compressive modulus and the permeability of the Schwarz P-based scaffold are high and the shear modulus is low, making it potentially suitable for the scenario where a high tension/compression is required, such as the spinal cage. The Diamond, the Gyroid, the Fischer-Koch S and the F-RD-based scaffolds have high shear moduli, but relatively low compressive moduli, making them generally suitable for the applications necessitating energy absorption rather than compression or tension stiffness [54]. The Fischer-Koch S-based scaffold has a high surface-to-volume ratio, but a low permeability because of the curved microstructures. Therefore, the Fischer-Koch S topology may be the most favorable one in the scenarios where nutrient is not limiting, e.g., in the application of bone fusion. It should also be noted that an ideal scaffold is the one which has the mechanical and biological properties similar to those of the replaced natural tissues [55, 56]. Therefore, optimizing and tuning the microstructure of scaffolds to mimic the behavior of the natural bone is the ultimate goal in scaffold design. It should be noted that the Young's modulus of the scaffold designed in the present study can be tuned from approximately 5.50 GPa to 33.00 GPa, which make them a good candidate for mimicking the mechanical properties of cortical bone (the modulus is ranged from approximately 5.00 GPa to 20.00 GPa) [57]. However, for mimicking the mechanical properties of trabecular bone (the modulus is ranged from approximately 0.15 GPa– 1.65 GPa) [57], the Ti-6Al-4V scaffold is too stiff and scaffolds made from other materials such as polymer should be used.

Regarding the comparison between the properties of the TPMS-based scaffolds and those of the traditional scaffolds, it is revealed in the present study that at the same porosity, the mechanical and permeability properties of the TPMS-based scaffolds are not always higher to those of the traditional scaffolds. For example, the compressive modulus of the Cube-based scaffold are the highest among all the scaffolds investigated, which could be due to the reason that the Cube-based scaffold has the highest proportion of beams aligned in the compressive loading direction. However, it should be noted that because there is no diagonal element in the Cube scaffold, its ability to resist the shear force is low and consequently the shear modulus of the Cube-based scaffold is the lowest. Additionally, it is revealed that the permeability of the Cube-based scaffold is the highest, the reason for which could be that the Cube-based scaffold has relatively straight and smooth surfaces. For clinical applications, the Cube topology is potentially favorable in the scenarios where the shear behavior is not limiting and the interconnective pores and compressive modulus are desired, for example, in the application of spinal cage. It should be noted that although at the same porosity, the compressive modulus and permeability of the Cube-based scaffold are higher than those of the TPMS-based scaffolds. The TPMS-based scaffolds have a high surface-to-volume ratio and an average surface curvature of zero, which could potentially facilitate the tissue regeneration [18, 58]. However, whether the biological behavior of the TPMS-based scaffolds is markedly superior to that of the traditional scaffolds and how to find a compromise between the mechanical (compression, shear, etc.) and biological (tissue regeneration, etc.) behaviors still need further investigations.

Regarding the relationship between the scaffold shear and compressive moduli, it is revealed that the normalized shear modulus is linearly correlated with the normalized compressive modulus with the slopes of all the regression lines bigger than one, implying that for each type of scaffold, the change in the relative shear modulus is always bigger than that in the relative compressive modulus. The slope for the Diamond-based scaffold is the highest,

implying that it is more effective to tune the shear modulus of the Diamond-based scaffold than tuning its compressive modulus. The fact that all the values except those for the Cube-based scaffold lies in the upper side of the diagonal line means that for the same scaffold, the normalized shear modulus is always higher than the normalized compressive modulus. Therefore, when the scaffold porosity increased, relative to the properties of the base material (compressive and shear moduli), the effective compressive modulus of the scaffold is reduced more than the reduction in the effective shear modulus for most scaffolds. This phenomenon is most obvious for the Diamond-based scaffold, implying that the Diamond-based scaffold is suitable for the scenario where a relative high shear modulus and a relative low compressive modulus are needed. It should be noted that when the scaffold is implanted into the long bone (e.g., femur), the scaffold is under the combined loading of axial compression and shear, due to the fact the femur is tilted approximately 7 degrees under the in vivo loading scenario [59]. The analysis on the compression and shear moduli of the scaffold could help derive the Zener anisotropy factor and understand the anisotropic mechanical behavior of the scaffold under the complex clinic loading scenario.

It should be noted that in the present study the scaffold permeability is used to reflect the biological behavior of scaffold, because it is revealed in previous studies that the scaffold permeability has a direct effect on the cell bioactivity, and a permeable scaffold allows for the efficient nutrient and oxygen diffusion and waste emission through its channels [60, 61]. It also should be noted that the Darcy's law is based on the CFD analysis where the laminar flow is assumed, while the Kozeny-Carman's relation is an empirical one. Because no permeability test is performed in the present study, no calibration can be done for the numerically calculated permeability, which could be reason the permeability from the two methods significantly differ in the high porosity region. Additionally, it should be noted that although the mechanical properties of scaffolds were normalized to the modulus of the base materials, the Poisson's ratio was fixed at that of the Ti-6Al-4V (i.e., 0.34), which may prevent the extension of the results to the base material with a different Poisson's ratio, e.g., the polymer. Therefore, in the future, the effect of the Poisson's ratio of the base material on the scaffold properties should also be investigated in order to understand the behaviors of scaffolds made from biodegradable polymers and etc. [62–64].

Several limitations related to the present study need to be discussed. First, only the elastic behaviors of the scaffolds are investigated and the nonlinear behaviors, such as the strength and the fatigue life, are not investigated. Indeed, the fatigue behavior is an important parameter reflecting the life expectation of the scaffolds. However, the elastic modulus is also an important parameter in the scaffold design because of its role in the load-bearing function [65], i.e., an excessively high elastic modulus can cause the undesirable stress-shielding phenomenon [66]. Furthermore, the mechanical environment (i.e., the distribution of compressive and shear moduli) plays an important role in the cell activities within scaffolds, such as cell proliferation and differentiation [67, 68]. Second, only the scaffold type of TPMS network solid is investigated. The TPMS sheet solids have been recently suggested as scaffold designs and showed significant potential benefits for tissue engineering [19, 69–71]. Recent studies [72] showed that the TPMS network solid and sheet solid have very dissimilar properties. Therefore, the investigation on the TPMS sheet solids still needs to be performed in the future. Last but not the least, the influence of scaffold microstructure on the permeability is investigated using only one set of parameters (flow rate, viscosity, etc.). Different scaffold microstructures may have a different influence on the pressure drop, and consequently the permeability calculated from the Darcy's law may change differently when the flow rate, the scaffold length and cross-section are changed. Therefore, in the future, the correlation between the scaffold microstructure and permeability should be investigated using more sets of data.

## Conclusion

In conclusion, the experimental and numerical approaches have been utilized to systematically reveal the underlying relationship between the scaffold properties and its microstructures. The main conclusions are as below:

- The bending dominated scaffolds (e.g., Diamond, Gyroid, Schwarz P, Fischer-Koch S and R-RD) tend to have a higher effective shear modulus. The stretching dominated scaffolds (e.g. Schwarz P, Cube, FD-Cube and Octa) tend to have a higher effective compressive modulus.

- The relative shear modulus of the scaffold changes faster than the relative compressive modulus, i.e., when the same amount of change in the scaffold porosity is made, the corresponding change in the relative shear modulus is larger than that in the relative compressive modulus.

- The permeability of the scaffold depends on the arrangement of the underlying microstructure, e.g., the structures with the simple and straight pores (e.g., Cube) have a higher permeability than the structures with the complex pores (e.g., Fischer-Koch S).

Some guidelines on the design of bone scaffolds are provided in the present study, for examples, the Fischer-Koch S topology is the most favorable one in the scenario where nutrient is not limiting, and the Cube topology is potentially favorable in the scenario where the shear behavior is not limiting.

## Author Contributions

**Conceptualization:** Yongtao Lu, Hanxing Zhu.

**Data curation:** Yongtao Lu, LiangLiang Cheng.

**Formal analysis:** Yongtao Lu, Zhuoyue Yang, Hanxing Zhu.

**Funding acquisition:** Yongtao Lu, Junyan Li, Hanxing Zhu.

**Investigation:** Yongtao Lu, Zhuoyue Yang.

**Methodology:** Yongtao Lu, Zhuoyue Yang.

**Project administration:** Yongtao Lu.

**Resources:** Yongtao Lu, LiangLiang Cheng.

**Software:** Zhuoyue Yang.

**Supervision:** Yongtao Lu.

**Validation:** Yongtao Lu, Zhuoyue Yang.

**Visualization:** Zhuoyue Yang.

**Writing – original draft:** Yongtao Lu.

**Writing – review & editing:** Yongtao Lu, LiangLiang Cheng, Junyan Li, Hanxing Zhu.

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
