## [Decision Letter · Decision Letter 0]

22 Jun 2020

PONE-D-20-13348

Relationship between the morphological, mechanical and permeability properties of porous bone scaffolds and the underlying microstructure

PLOS ONE

Dear Dr. Lu,

Thank you for submitting your manuscript to PLOS ONE. After careful consideration, we feel that it has merit but does not fully meet PLOS ONE’s publication criteria as it currently stands. Therefore, we invite you to submit a revised version of the manuscript that addresses the points raised during the review process.

We look forward to receiving your revised manuscript.

Kind regards,

Yanyu Chen

Academic Editor

PLOS ONE

Journal Requirements:

Additional Editor Comments (if provided):

Reviewers' comments:

Reviewer's Responses to Questions

**Comments to the Author**

1. Is the manuscript technically sound, and do the data support the conclusions?

Reviewer #1: No

Reviewer #2: Yes

Reviewer #3: Yes

Reviewer #4: Partly

2. Has the statistical analysis been performed appropriately and rigorously? 

Reviewer #1: N/A

Reviewer #2: Yes

Reviewer #3: No

Reviewer #4: No

3. Have the authors made all data underlying the findings in their manuscript fully available?

Reviewer #1: No

Reviewer #2: Yes

Reviewer #3: Yes

Reviewer #4: No

4. Is the manuscript presented in an intelligible fashion and written in standard English?

Reviewer #1: No

Reviewer #2: Yes

Reviewer #3: Yes

Reviewer #4: Yes

5. Review Comments to the Author

Reviewer #1: The current study investigates the relationship between topology, mechanical properties and permeability of triply periodic minimal surfaces (TPMS) based lattices and other strut-based lattices. The work follows a finite elements analyses framework and experimental validation of the mechanical part.

Unfortunately, this paper is weakly written, and does not provide any new significant findings at all. therefore, it is not recommended for publication.

Introduction

The introduction seems to be missing lots of relevant works that are hard to miss and reported the properties of TPMS-based materials. For example, the work of Montazerian et al (10.1016/j.matdes.2017.04.009) reported the elastic properties and permeability of a wide range of TPMS-based materials including those reported in this work. The study of Kapfer et al (10.1016/j.biomaterials.2011.06.012) which dates back to 2011 also investigates the elastic and permeability properties. Abueidda et al. (10.1016/j.mechmat.2016.01.004) also reported the elastic properties of TPMS-based materials numerically. These are only few examples.

The authors claim that literature is missing studies that compare the properties of TPMS based materials with other lattice types. This is also not true as the work of Al-Ketan and coworkers (10.1016/j.addma.2017.12.006), (10.1002/adem.201800029), and (10.1557/jmr.2018.1) have extensively discussed the difference in mechanical behavior of TPMS-based materials in comparison with strut-based materials.

Results and discussion

Unfortunately, the presented results seem to be missing a lot, the authors claim that they performed a mesh convergence study without reporting the results of this study, or the criteria used to decide on the mesh size. The authors also did not show any stress contours!!

The authors also claim to have validated the mechanical properties experimentally. However, the authors did not show any figure of the 3D printed samples, or if the actual relative density matches that of the designed. The authors also did not present a single stress-strain response or the deformation pattern of the different lattices.

On page 16, the authors state “However, it should be noted that, because the TPMS sheet solids have the same microstructure topology as their network solid counterparts, the features of the mechanical and permeability properties of the TPMS sheet solids should be similar to their network solid counterparts. However, this needs to be confirmed in the future studies.”

This statement is very wrong and the properties should not be similar. In fact, several studies have already shown that solid-networks and sheet-networks have very dissimilar properties. For example, Kapfer et al (10.1016/j.biomaterials.2011.06.012) and Al-Ketan et al (10.1002/adem.201800029) among others. In fact, a recent review by Al-Ketan et al ( 10.1002/adem.201900524) discusses in detail the difference between sheet-based and network-based lattices.

In conclusion, this work is significantly missing a lot of proper analysis, data presentation, and comprehensive discussion in light of the ubiquitous studies presented to data with respect to TPMS-based materials. this reviewer does not recommend this work for publication.

Reviewer #2: This is an interesting work. The relationships between geometrical configuration of lattice cell, mechanical properties and permeability were comprehensively discussed through finite element method. Some issues should be addressed before considering this paper for publication.

(1) Please highlight the main significance of this work in introduction. The authors need to highlight the novelty of this work clearly.

(2) The introduction and reference sections should be enhanced and improved with up-to-date published works, for example

Jia et al. An experimental and numerical investigation of compressive response of designed Schwarz Primitive triply periodic minimal surface with non-uniform shell thickness. Extreme Mechanics Letters 37 (2020) 100671.

Li et al. Architecture design of periodic truss-lattice cells for additive manufacturing. Additive Manufacturing 34 (2020) 101172.

(3) The authors mentioned that a mesh convergence study was performed for the FEA. Please add more details about the convergence analysis.

(4) The authors mentioned that compressive tests were carried out, so please add the test curves and failure morphologies of samples, and compared with the FEA results.

In all, I think this paper should be minor revised before publication.

Reviewer #3: The paper presents results which are both novel and useful, especially for the designer of scaffold structures for a range of mechanical and fluid-flow applications. The range of lattice types examined, and the use of CFD to determine scaffold permeability, mean this paper could be a valuable contribution to the field.

However, it is somewhat confusing to see quadratic fits applied to the property-porosity relationships (shown in table 2), as well as Gibson-Ashby type power laws (in table 3). In the field of cellular structures, the latter is far more commonly used, and indeed is supported by the analytical solutions for structural deformation summarised in Gibson and Ashby's book, Cellular Solids. The authors have not justified their use of quadratic models here. This is not satisfactory, since the application of a particular mathematical model to experimental data must always be justified according to the physical phenomena. In this paper, this applies to the analysis of the compressive modulus, shear modulus and permeability, the fitting for which should be revised, supported with good reasons, or both. I will be happy to review the manuscript again if the authors choose to make these revisions.

As a final note, in table 2 and throughout the paper, it is not appropriate to use the terms y and x when describing other properties. These should be changed for the actual properties in question; e.g., E = rho^n, not y=x^n.

Reviewer #4: The paper entitled "Relationship between the morphological, mechanical and permeability properties of porous bone scaffolds and the underlying microstructure" is focused on the characterization of mechanical properties and permeability for the purpose of designing bone scaffolds mainly based on computational tools followed by limited experimental validation. In general, there are some concerns as detailed below that put the paper's suitability for publication under question:

General comments:

-Although the topic of design of a porous scaffold is yet evolving, I could not find any of the results really adding to the current state-of-the-art or improving the current design approaches. The properties of uniform TPMS and lattice unit cells are extensively studied and well-documented. Currently, the research is shifting to address more advanced questions related to areas such as functionally gradient scaffolds and structures with more complex geometries other than cube or cylinder samples similar to organs and organ shaped implants. Therefore I would question the lack of novelty of the paper.

-Why the characterization of shear modulus is important in the design of scaffold? Please provide an example where a scaffold undergoes shear load in the body to justify.

-The paper should include a conclusion section.

Abstract

-p. 2: Abstract- the findings highlighted in the abstract are obvious evidence that have been demonstrated extensively before and therefore does not seem interesting the readers in the field (for instance, the deformation mechanism bending/stretching dominated for the mentioned geometries, the high surface area at 0.5 porosity).

-p. 2 l. 19: What do the authors mean by: "For all the scaffolds, it is more effective to tune the relative shear modulus than tuning the relative compressive modulus" and why this finding is important for the design of scaffolds?

Introduction:

-p. 3 l. 27: Unclear what the term "conductivity" refers to (thermal, electrical)? Besides, how is this conductivity important making it attractive for scaffold design?

Materials and Methods:

-p. 5: Section "The finite element models of bone scaffolds"- The material model used for the simulation is missing. Is it a perfectly plastic model? if so, please report the yield stress, etc. Please also mention the number of meshes representing your FE model.

-p. 5 l. 8: The authors mention that they used K3D surf software to define the models. What is the maximum number of unit cell sizes that can be modeled, meshed, and simulated this way? Furthermore, the TPMS equations used to model the constructs are missing. Please also describe in detail how different porosity values were obtained.

-p. 5 l. 17: Why the void phase needed to be meshed in order to define periodic boundary condition? Can one define it on the solid phase only?

-p. 5 l. 21: Since the paper focuses on the computational design of the scaffold, please also provide the results of your convergence study. Figure 1c does not present anything regarding mesh convergence study that is cited here.

-p. 6 l. 16: What do the authors mean by "nominal volume"? Please clarify if the S/V is defined by the volume of the cube encompassing the scaffold or the volume of the solid phase.

-p. 6 l. 28: Why six different individual loading is defined when the unit cells are symmetric?

-p. 7 l. 20: An inlet velocity of 0.1 mm/s is defined at the scaffold inlet. Fluid flow is not developed at the proximity of the channel inlet in a CFD model. Does the fact that scaffold falls into that transitioning region affect the accuracy of the results? Besides, Darcy's equation is valid for the case of laminar flow. Have the authors investigated if this is not an issue in their model by looking at the effect of flow rate around 0.1 mm/s on permeability value?

- Section "Validation of the predictions of the FE models of scaffolds": Many details on the experimental procedure are missing. Please explain the printing parameters such as scanning speed, SLM laser power, etc. Also it is important to demonstrate a figure showing the images of the fabricated parts. For the mechanical test, what is the loading rate?

Results

-I would discourage repeating the numbers for the results in the text when they are presented in the figures (see page 10-11)

-The authors argue in the introduction that increasing porosity despite increasing permeability, reduces the scaffold strength but they don't touch on mechanical strength throughout the experiments/simulations. The compressive strength seems to be worth much more attention than e.g. shear modulus.

-Figure 3(c): What is special about the relation between compressive modulus and shear modulus? In what capacity is this important for the scaffold design?

-Table 1-3: The number of meaningful digits (digits after the decimals) should be consistent in each table.

-Table 1: Please provide a % deviation of FE from experiments. The discrepancies between FE and experiments seem much less than the typical order in the literature. Can the authors provide their deviation with the literature?

-The stress-strain curves for both experimental compression tests and FE simulation must be presented and compared.

-Table 2: Table caption is wrong. The table doesn't represent the relationship between mechanical properties, surface-to-volume ratio and permeability. Rather they show the relation between these parameters and porosity. Besides, the regression should be defined separately for each curve fitting.

-Table 3: Why E0 is "-" for Cube, FD-Cube and Octa?

Discussion:

-p. 13, l. 5-9: References 45 and 46 have not characterized the experimental/numerical permeability. Besides, the reported comparisons of computational and experimental permeability in the literature are highly significant with a factor of ~0.1 (see DOI: 10.1016/j.jbiomech.2012.01.019 10.1016/j.matdes.2017.03.006)

-p. 13, l. 9-11: The largest order of permeability in the cited papers (Ref. 47, 48), as well as the author's results (Fig. 4(b)), starts from 10^-9 (not 10^-8)

-p. 15, l. 29: do we have "gas diffusion" in a scaffold?

-p. 16, l. 15: Given the authors are mentioning the stress-shielding effect, please compare the results obtained in this study with the natural properties of bone to elucidate how far are we from mimicking the properties of native bone.

-p. 16, l. 21: "it should be noted that, because the TPMS sheet solids have the same microstructure topology as their network solid counterparts, the features of the mechanical and permeability properties of the TPMS sheet solids should be similar to their network solid counterparts. However, this needs to be confirmed in future studies": It is very unclear what the authors are trying to convey.

-Fig. 4(c): can you please discuss what is the reason for the huge difference between the permeability from CFD and Kozeny-Carman model and comment on which one is a more reliable model to take into account when designing scaffolds for permeability?

-In practice, the scaffold is placed and fit in the holes drilled in bone by the surgeon. Imagine as if a cylinder is mechanically loaded in transverse direction. Given the cubic models studied here, can you comment on how can these results can be translated and applied to the case of complex loading configurations that are applied under the physiological conditions?

-Does the simulation parameters such as viscosity, flow rate, scaffold length, and cross-section affect computational permeability? If so, why the permeability can be a suitable parameter for correlating pore shape to biological behavior while it is cross-sensitive to the abovementioned factors?

-The deformation mechanism is discussed in the context of mechanical properties. Please discuss what is the implications of deformation mechanisms on biological performance?

6. PLOS authors have the option to publish the peer review history of their article (what does this mean?). If published, this will include your full peer review and any attached files.

Reviewer #1: Yes: Oraib Al-Ketan

Reviewer #2: No

Reviewer #3: No

Reviewer #4: No

---

## [Author Response · Author response to Decision Letter 0]

29 Jul 2020

Replies to reviewers’ comments

Manuscript Number: PONE-D-20-13348 

Title: Relationship between the morphological, mechanical and permeability properties of porous bone scaffolds and the underlying microstructure 

Thanks to the reviewers for their valuable suggestions. They have contributed to improve the quality of the paper. We hope the responses we provide below will answer their concerns and shed light on unclear parts of the study.

Editor’s recommendation 

If applicable, we recommend that you deposit your laboratory protocols in protocols.io to enhance the reproducibility of your results. Protocols.io assigns your protocol its own identifier (DOI) so that it can be cited independently in the future.

Replies: we have uploaded the model data into a public repository ( Figshare) , which can be assessed at https://doi.org/10.6084/m9.figshare.12721325

Reviewer #1: 

The current study investigates the relationship between topology, mechanical properties and permeability of triply periodic minimal surfaces (TPMS) based lattices and other strut-based lattices. The work follows a finite elements analyses framework and experimental validation of the mechanical part.

Unfortunately, this paper is weakly written, and does not provide any new significant findings at all. Therefore, it is not recommended for publication.

Introduction

The introduction seems to be missing lots of relevant works that are hard to miss and reported the properties of TPMS-based materials. For example, the work of Montazerian et al (10.1016/j.matdes.2017.04.009) reported the elastic properties and permeability of a wide range of TPMS-based materials including those reported in this work. The study of Kapfer et al (10.1016/j.biomaterials.2011.06.012) which dates back to 2011 also investigates the elastic and permeability properties. Abueidda et al. (10.1016/j.mechmat.2016.01.004) also reported the elastic properties of TPMS-based materials numerically. These are only few examples.

Replies: Thanks for the comment. The recommended papers (10.1016/j.matdes.2017.04.009; 10.1016/j.biomaterials.2011.06.012; 10.1016/j.mechmat.2016.01.004) have been included into the manuscript. Indeed, the elastic and permeability properties of TPMS-based scaffolds have been investigated in the previous studies. However, a systematical investigation on the relationship between the scaffold properties and its underlying microstructure, i.e., how the scaffold performance is influenced by its microstructure design, has not been performed. In the present study, some novel findings, such as the stretching-dominated structures have higher effective compressive modulus and the scaffold porosity will influence the scaffold relative shear modulus more than the relative compressive modulus, have not been reported in the previous studies. We have rephrased the sentences in the manuscript to highlight the contributions/novelty of the present study. Some of the rephrased sentences are as below:

“It is revealed that the surface-to-volume ratio of the Fischer-Koch S-based scaffold is the highest among the scaffolds investigated. The mechanical analysis revealed that the bending deformation dominated structures (e.g., the Diamond, the Gyroid, the Schwarz P) have higher effective shear moduli. The stretching deformation dominated structures (e.g., the Schwarz P, the Cube) have higher effective compressive moduli. For all the scaffolds, when the scaffold porosity is changed the same amount, the associated change in the scaffold relative shear modulus is larger than that in the relative compressive modulus.” (in the Abstract part)

“A comprehensive comparison of the morphological, mechanical and permeability properties of different TPMS-based scaffolds is still missing and the mechanism of the microstructure-driven scaffold properties is still unclear.” (in the Introduction part, Page 4 Lines 23 - 26)

“For all the scaffolds, when the same amount of change in scaffold porosity is made, the associated change in the scaffold relative shear modulus is larger than that in the relative compressive modulus.” (in the Conclusion part, Page 17 Lines 23 - 25)

The authors claim that literature is missing studies that compare the properties of TPMS based materials with other lattice types. This is also not true as the work of Al-Ketan and coworkers (10.1016/j.addma.2017.12.006), (10.1002/adem.201800029), and (10.1557/jmr.2018.1) have extensively discussed the difference in mechanical behavior of TPMS-based materials in comparison with strut-based materials.

Replies: Thanks for the recommendation of relevant papers. What we meant is that the comparison of the microstructure-driven performance between the TPMS based with other lattice types were missing in the literature. We have now rephrased the sentences and added the recommended papers into the manuscript. 

“A comprehensive comparison of the morphological, mechanical and permeability properties of different TPMS-based scaffolds is still missing and the mechanism of the microstructure-driven scaffold properties is still unclear.” (Page 4 Lines 23 - 26)

Results and discussion

Unfortunately, the presented results seem to be missing a lot, the authors claim that they performed a mesh convergence study without reporting the results of this study, or the criteria used to decide on the mesh size. The authors also did not show any stress contours!!

Replies: We have added the details as suggested. Regarding the mesh convergence, the FE mesh size was refined until the influence of the mesh size on the FE predictions (the normalized compressive and shear moduli) was less than 0.5%. We have also added a figure (Fig 3) to demonstrate the mesh convergence study. Regarding the stress contours, unlike the paper (10.1002/adem.201800029), the focus of the present study is on the effective elastic behavior of the scaffolds, and no plastic or failure behaviors of the scaffolds are involved. So the stress contours does not make any contribution to the conclusion in the present study. Thanks for the suggestion anyway. 

“When calculating the mechanical properties, a mesh convergence study was performed to ensure that the influence of the mesh size on the FE predicted compressive and shear moduli was less than 0.5% (regarded as converged) (Fig 3)…” (Page 7 Lines 17 - 20)

The authors also claim to have validated the mechanical properties experimentally. However, the authors did not show any figure of the 3D printed samples, or if the actual relative density matches that of the designed. The authors also did not present a single stress-strain response or the deformation pattern of the different lattices.

Replies: Following the suggestion, we have now added one representative 3D printed scaffold (Figure 1f), one representative stress-strain curve from the mechanical testing (Figure 4). Indeed, there could be a large discrepancy between the designed and AM produced scaffold. Therefore, to minimize the influence of AM errors, we checked and controlled the quality of the AM produced scaffold. The scaffolds with a big error in the porosity were not used for the FE validation. We added the following clarification in the manuscript:

“The defected scaffolds, i.e. the deviation of the porosity from the design value is larger than 5%, were disposed and five samples per scaffold porosity were selected.” (Page 9 Lines 23 - 25)

On page 16, the authors state “However, it should be noted that, because the TPMS sheet solids have the same microstructure topology as their network solid counterparts, the features of the mechanical and permeability properties of the TPMS sheet solids should be similar to their network solid counterparts. However, this needs to be confirmed in the future studies.” This statement is very wrong and the properties should not be similar. In fact, several studies have already shown that solid-networks and sheet-networks have very dissimilar properties. For example, Kapfer et al (10.1016/j.biomaterials.2011.06.012) and Al-Ketan et al (10.1002/adem.201800029) among others. In fact, a recent review by Al-Ketan et al ( 10.1002/adem.201900524) discusses in detail the difference between sheet-based and network-based lattices.

Replies: Thanks for the comments. We have read the recommended papers and modified the statements in the relevant place:

“Recent studies [72] showed that the TPMS network solid and sheet solid have very dissimilar properties. Therefore, the investigation on the TPMS sheet solids still needs to be performed in the future.” (Page 17 Lines 3 - 6)

Reviewer #2: 

This is an interesting work. The relationships between geometrical configuration of lattice cell, mechanical properties and permeability were comprehensively discussed through finite element method. Some issues should be addressed before considering this paper for publication.

(1) Please highlight the main significance of this work in introduction. The authors need to highlight the novelty of this work clearly.

Replies: The main contribution of the present study is that the mechanism between scaffold properties and the underlying microstructure was revealed and consequently some useful findings (e.g., for the scaffolds, it is more effective to tune the relative shear modulus than tuning the relative compressive modulus) can be used when designing bone scaffolds. We have now highlighted the main significance of the work in the Abstract (the highlighted part). 

(2) The introduction and reference sections should be enhanced and improved with up-to-date published works, for example

Jia et al. An experimental and numerical investigation of compressive response of designed Schwarz Primitive triply periodic minimal surface with non-uniform shell thickness. Extreme Mechanics Letters 37 (2020) 100671.

Li et al. Architecture design of periodic truss-lattice cells for additive manufacturing. Additive Manufacturing 34 (2020) 101172.

Replies: Thanks for the recommendation. We have included the up-to-date papers into the manuscript; please see the added references of 27 and 28. 

(3) The authors mentioned that a mesh convergence study was performed for the FEA. Please add more details about the convergence analysis.

Replies: The FE mesh size was refined until the influence of the mesh size on the FE predictions (the compressive and shear moduli) was less than 0.5%. We have added the clarification in the manuscript as below:

“When calculating the mechanical properties, a mesh convergence study was performed to ensure that the influence of the mesh size on the FE predicted compressive and shear moduli was less than 0.5% (regarded as converged) (Fig 3)…” (Page 7 Lines 17 - 20)

(4) The authors mentioned that compressive tests were carried out, so please add the test curves and failure morphologies of samples, and compared with the FEA results.

Replies: The failure behavior of the samples is not the scope of the present study and not simulated and investigated in the present study. The failure is neither simulated in the FE analysis. We have added a representative stress-strain curve from the mechanical testing (see Figure 4), and made the clarification in the manuscript:

“…and no nonlinear mechanical properties were defined because of the linear elastic simulation.” (Page 6 Lines 25 - 26)

In all, I think this paper should be minor revised before publication.

Reviewer #3: 

The paper presents results which are both novel and useful, especially for the designer of scaffold structures for a range of mechanical and fluid-flow applications. The range of lattice types examined, and the use of CFD to determine scaffold permeability, mean this paper could be a valuable contribution to the field.

Replies: Thanks for the comment. We appreciate the suggestions for improving the quality of the present study. 

However, it is somewhat confusing to see quadratic fits applied to the property-porosity relationships (shown in table 2), as well as Gibson-Ashby type power laws (in table 3). In the field of cellular structures, the latter is far more commonly used, and indeed is supported by the analytical solutions for structural deformation summarised in Gibson and Ashby's book, Cellular Solids. The authors have not justified their use of quadratic models here. This is not satisfactory, since the application of a particular mathematical model to experimental data must always be justified according to the physical phenomena. In this paper, this applies to the analysis of the compressive modulus, shear modulus and permeability, the fitting for which should be revised, supported with good reasons, or both. I will be happy to review the manuscript again if the authors choose to make these revisions.

Replies: Thanks for the comment. Indeed, the semi-empirical formula of Gibson and Ashby is widely used to describe the elastic modulus of scaffold and its density, but the relation is not a good model for the scaffold permeability. In the present study, the aim of using quadratic models was to make the comparison with the permeability-porosity relationship. Additionally, the Gibson and Ashby’s fitting was also performed to reveal the underlying physical phenomena. Please see the clarification added in the manuscript: 

“To make comparisons among different properties, the statistical regression equations (quadratic or other forms) and the coefficients of determination (R2) were computed for the relationships between the scaffold mechanical properties and the scaffold porosity and between the scaffold permeability and the porosity.” (Page 8 Lines 24 - 28)

“Additionally, to reveal the underlying physical phenomena, the relationship between the relative elastic compressive modulus and the scaffold volume fraction were described using the exponential function proposed by Gibson and Ashby [31, 32]” (Page 8 Lines 28 – 29 and Page 9 Lines 1 -2)

As a final note, in table 2 and throughout the paper, it is not appropriate to use the terms y and x when describing other properties. These should be changed for the actual properties in question; e.g., E = rho^n, not y=x^n.

Replies: Thanks for the suggestion. We have changed the corresponding expressions throughout the paper. 

Reviewer #4: 

The paper entitled "Relationship between the morphological, mechanical and permeability properties of porous bone scaffolds and the underlying microstructure" is focused on the characterization of mechanical properties and permeability for the purpose of designing bone scaffolds mainly based on computational tools followed by limited experimental validation. In general, there are some concerns as detailed below that put the paper's suitability for publication under question: 

General comments:

-Although the topic of design of a porous scaffold is yet evolving, I could not find any of the results really adding to the current state-of-the-art or improving the current design approaches. The properties of uniform TPMS and lattice unit cells are extensively studied and well-documented. Currently, the research is shifting to address more advanced questions related to areas such as functionally gradient scaffolds and structures with more complex geometries other than cube or cylinder samples similar to organs and organ shaped implants. Therefore I would question the lack of novelty of the paper.

Replies: Thanks for the comment. Indeed, the up-to-date research has shifted to the functionally gradient scaffold, the bionic scaffold, the smart porous structures, etc. However, the mechanism of the microstructure-driven performance of the bone scaffolds is still unclear. Therefore, the main contribution (novelty) of the present paper is that the relationship between the scaffold microstructure and its properties is systematically investigated. We have highlighted the contribution (novelty) of the present paper in the manuscript:

“However, many bone scaffold microstructure topologies exist and it is still unclear which topology to use when designing scaffold for a specific application. The aim of the present study was to reveal the mechanism of the microstructure-driven performance of bone scaffold and thus to provide guideline on scaffold design.” (In the Abstract)

“The main contribution of the present study is that the relationship between scaffold properties and the underlying microstructure is systematically investigated and thus some guidelines on the design of bone scaffolds are provided, for example, in the case when a high surface-to-volume ratio is required, it is suggested to use the Fischer-Koch S based scaffold.” (In the Abstract)

-Why the characterization of shear modulus is important in the design of scaffold? Please provide an example where a scaffold undergoes shear load in the body to justify.

Replies: Thanks for the nice comment. When the scaffold is implanted into the long bone (femur), the scaffold is under the combined loadings of axial compression and shear, because the femur is tilted approximately 7 degrees under the in vivo scenario (Entezari et al., 2019). Any complex loading scenario can be converted to pure axial compression and pure shear. Therefore, the characterization of shear modulus can help understand the mechanical behavior of scaffold under complex clinic loading scenario. The clarification has been made in the manuscript as below: 

“It should be noted that when the scaffold is implanted into the long bone (e.g., femur), the scaffold is under the combined loading of axial compression and shear, due to the fact the femur is tilted approximately 7 degrees under the in vivo loading scenario [59]. The analysis on the compression and shear moduli of the scaffold could help understand the anisotropic mechanical behavior of the scaffold under the complex clinic loading scenario.” (Page 15 Lines 27 – 29 and Page 16 Lines 1 - 4)

-The paper should include a conclusion section.

Replies: Thanks for the suggestion. We have now added a conclusion section:

“In conclusion, experimental and numerical approaches have been utilized to systematically reveal the underlying relationship between the scaffold properties and its microstructures. The main conclusions are as below: …” (Page 17 Lines 15 - 17)

Abstract

-p. 2: Abstract- the findings, highlighted in the abstract, are obvious evidence that have been demonstrated extensively before and therefore does not seem interesting the readers in the field (for instance, the deformation mechanism bending/stretching dominated for the mentioned geometries, the high surface area at 0.5 porosity).

Replies: Thanks for the comment. We have modified the sentences in the abstract to put the emphasis on the relationship between the scaffold microstructure and its properties, which is unclear in the literature. 

‘It is revealed that the surface-to-volume ratio of the Fischer-Koch S-based scaffold is the highest among the scaffolds investigated. The mechanical analysis revealed that the bending deformation dominated structures (e.g., the Diamond, the Gyroid, the Schwarz P) have higher effective shear moduli. The stretching deformation dominated structures (e.g., the Schwarz P, the Cube) have higher effective compressive moduli.’ (in the Abstract )

-p. 2 l. 19: What do the authors mean by: "For all the scaffolds, it is more effective to tune the relative shear modulus than tuning the relative compressive modulus" and why this finding is important for the design of scaffolds?

Replies: Sorry for the confusion. What we meant is that when the same amount of change is made in the scaffold porosity, the amount of change in the scaffold relative shear modulus is larger than that in the relative compressive modulus. This finding implies that when switching from the high porosity scaffold to the low porosity scaffold, the change in the scaffold relative shear modulus will be larger than that in the relative compressive modulus and consequently the Zener anisotropy factor will be influenced. This message can help understand the evolution of the scaffold mechanical anisotropic property and thus help the design of scaffold. We have made the relevant changes in the Abstract and Discussion parts as below: 

“For all the scaffolds, when the scaffold porosity is changed the same amount, the associated change in the scaffold relative shear modulus is larger than that in the relative compressive modulus.” (in the Abstract)

“The analysis on the compression and shear moduli of the scaffold could help derive the Zener anisotropy factor and understand the anisotropic mechanical behavior of the scaffold under the complex clinic loading scenario.” (Page 16 Lines 1 - 4)

Introduction:

-p. 3 l. 27: Unclear what the term "conductivity" refers to (thermal, electrical)? Besides, how is this conductivity important making it attractive for scaffold design?

Replies: The term “conductivity” refers to the thermal and electrical conductivity of the scaffold. Variable/tunable conductivity means that the scaffold conductivity can be made anatomic location-specific and subject-specific so as to better meet the physiological needs of human body when designing the scaffold. We have the clarification as below: 

“… a variable /tunable electrical/thermal conductivity [21], which can make their properties anatomical location-specific and subject-specific and consequently can largely increase their potentials in the applications in biomedicine and relevant fields [22]” (Page 3 Lines 27 -29)

Materials and Methods:

-p. 5: Section "The finite element models of bone scaffolds"- The material model used for the simulation is missing. Is it a perfectly plastic model? if so, please report the yield stress, etc. Please also mention the number of meshes representing your FE model.

Replies: Only the effective elastic properties were investigated in the manuscript, so no nonlinear behavior was involved and needed in the linear elastic simulation. After the mesh convergence study, element with the size of approximately 0.1 mm were generated for the unit cell model and element with the size of approximately 0.2 mm were generated for the 4×4×4 scaffold model. We have added the missing information into the manuscript. 

“Therefore, a Young’s modulus of 110.0 GPa [40] and a Poisson’s ratio of 0.34 were defined for the solid domain and no nonlinear mechanical properties were defined because of the linear elastic simulation.” (Page 6 Lines 25 -26)

“When calculating the mechanical properties, a mesh convergence study was performed to ensure that the influence of the mesh size on the FE predicted compressive and shear moduli was less than 0.5% (regarded as converged) (Fig 3) and the element size of approximately 0.1 mm was used, which resulted in approximately 0.06 million elements for the Fischer-Koch S unit cell model with the porosity of 0.55.” (Page 7 Lines 17 - 22)

“The void domains were meshed using the tetrahedral element with the size of approximately 0.2 mm, which resulted in approximately 0.64 million elements for the Fischer-Koch S scaffold with the porosity of 0.55.” (Page 7 Lines 28 -29)

-p. 5 l. 8: The authors mention that they used K3D surf software to define the models. What is the maximum number of unit cell sizes that can be modeled, meshed, and simulated this way? Furthermore, the TPMS equations used to model the constructs are missing. Please also describe in detail how different porosity values were obtained.

Replies: We used the unit (representative) cell technique, so one periodic representative model (unit cell model) with the dimension of 2.5 mm × 2.5 mm × 2.5 mm was used. Different porosity values of the scaffold were obtained by changing the constant in the TPMS nodal equations. Following the suggestion, we added the TPMS equations (please see the added Table 1) and the missing information. 

“The approximated periodic nodal equations for the five TPMS scaffolds are presented in Table 1 and different scaffold porosities can be obtained by changing the constant (C) in the TPMS nodal equations.” (Page 6 Lines 8 -10)

-p. 5 l. 17: Why the void phase needed to be meshed in order to define periodic boundary condition? Can one define it on the solid phase only?

Replies: This is a good point. In unit cell models of some scaffolds, the solid phase finishes at one surface and there is no corresponding at the opposite surface, in which case the definition of periodic boundary condition is not possible if there is no element present. Therefore, we generated the meshes also for void phase to facilitate the application of periodic boundary condition. 

“In the FE unit cell model, a Young’s modulus of 1.0 MPa and a Poisson’s ratio of 0.45 [41] were defined to facilitate the definition of periodic boundary condition, because in some unit cell models, the scaffold solid phase finished at one exterior surface and there were no corresponding elements in the opposite surface.” (Page 6 Lines 28 -29)

-p. 5 l. 21: Since the paper focuses on the computational design of the scaffold, please also provide the results of your convergence study. Figure 1c does not present anything regarding mesh convergence study that is cited here.

Replies: Following the suggestion, we added the curves for the mesh convergence study, please see the added Figure 3. The following sentences were added to describe the convergence study:

“When calculating the mechanical properties, a mesh convergence study was performed to ensure that the influence of the mesh size on the FE predicted compressive and shear moduli was less than 0.5% (regarded as converged) (Fig 3)…” (Page 7 Lines 17 - 20)

-p. 6 l. 16: What do the authors mean by "nominal volume"? Please clarify if the S/V is defined by the volume of the cube encompassing the scaffold or the volume of the solid phase.

Replies: Sorry for the confusion. The S/V is defined by the volume of the cube encompassing the scaffold. We have made the clarification in the manuscript as below:

“Scaffold porosity was calculated as the value using the scaffold void volume divided by its nominal volume (i.e., the volume of the cube encompassing the scaffold)” (Page 6 Lines 18 -19)

-p. 6 l. 28: Why six different individual loading is defined when the unit cells are symmetric?

Replies: It is demonstrated from one of our previous publications (Lu et al., JMBBM, 2019) that the Schwarz P and F-RD are cubic symmetric, and the Diamond, Gyroid and Fischer-Koch S are threefold rotational symmetric. However, all the scaffolds investigated have the three non-zero constants in the elasticity matrix. Given the fact that the calculation is efficient (also to be conservative), we defined the six individual loading to obtain the elasticity matrix. After the suggestion, we switched to three individual loading, the answers are the same to those obtained from the six individual loading. So we updated the text in the manuscript to make it clear: 

“Using the FE unit cell model, the effective elasticity tensor for each scaffold was first derived by solving the material constitutive equations, established by defining three individual loading, i.e., ε_x= 0.01, ε_y=0.01, ε_xy=0.01[42, 43]. The reason to define three individual loading is that all the scaffolds investigated have three nonzero constants in the elasticity matrix and can be regarded as the cubic symmetric structure [42]” (Page 7 Lines 1 - 6)

-p. 7 l. 20: An inlet velocity of 0.1 mm/s is defined at the scaffold inlet. Fluid flow is not developed at the proximity of the channel inlet in a CFD model. Does the fact that scaffold falls into that transitioning region affect the accuracy of the results? Besides, Darcy's equation is valid for the case of laminar flow. Have the authors investigated if this is not an issue in their model by looking at the effect of flow rate around 0.1 mm/s on permeability value?

Replies: Thanks for the comment. We have performed the sensitivity study and found using the 4×4×4 model for the CFD analysis can get the accurate results. Indeed the Darcy’s equation is valid for the laminar flow and we set-up the laminar flow simulation and have checked the 0.1 mm/s achieved the laminar flow. We have added the clarification in the manuscript: 

“A sensitivity study showed that the relative difference between the permeability calculated from the 4×4×4 and the 5×5×5 FE models was as small as 0.1% (i.e., the boundary effect has been removed).” (Page 6 Lines 1 -4)

“The fluid was modeled as incompressible water with a viscosity of 0.001 Pa and a density of 998.2 kg/m3 [45] and the case of laminar flow was simulated.” (Page 8 Line 8)

- Section "Validation of the predictions of the FE models of scaffolds": Many details on the experimental procedure are missing. Please explain the printing parameters such as scanning speed, SLM laser power, etc. Also it is important to demonstrate a figure showing the images of the fabricated parts. For the mechanical test, what is the loading rate?

Replies: The scanning speed is 0.04 m/s, the SLM laser power is 350.0W and the hatch angle is 90 degrees. We have added a figure (Fig 1f) to demonstrate the AM produced scaffold. For the mechanical test, the loading rate is 0.5 mm/min. We appreciated the suggestions and have added the required information in the manuscript as below: 

“The designed Schwarz P and Diamond-based scaffolds were produced using the additive manufacturing method of Selective Laser Melting (SLM) (Renishaw AM250, Renishaw plc., Gloucestershire, UK) with the scanning speed of 0.04 m/s, the laser power of 350.0 W and the hatch angle of 90 degrees.” (Page 9 Lines 22 -23)

“… the quasi-static testing was performed, where the crosshead speed was 0.5 mm/min.” (Page 9 Lines 27 -28)

Results

-I would discourage repeating the numbers for the results in the text when they are presented in the figures (see page 10-11)

Replies: we appreciate this comment and have modified the text in the Results part. 

-The authors argue in the introduction that increasing porosity despite increasing permeability, reduces the scaffold strength but they don't touch on mechanical strength throughout the experiments/simulations. The compressive strength seems to be worth much more attention than e.g. shear modulus.

Replies: Thanks for the comments. Indeed, the strength and fatigue are worthy much more attention than the effective elastic modulus. Regarding the strength and fatigue, it is more reliable to use the experimental technique than the computational simulation, because much is still unknown in the prediction of scaffold nonlinear behavior. The present study focused on the relation between the scaffold properties and porosity in various scaffolds using the numerical technique. The investigation on the nonlinear behavior is our ongoing work. We have discussed this point in the Limitation part of the manuscript as below:

“First, only the elastic behaviors of the scaffolds are investigated and the nonlinear behaviors, such as the strength and the fatigue life, are not investigated. Indeed, the fatigue behavior is an important parameter reflecting the life expectation of the scaffolds. However, the elastic modulus is also an important parameter in the scaffold design because of its role in the load-bearing function [65], i.e., an excessively high elastic modulus can cause the undesirable stress-shielding phenomenon [66]. Furthermore, the mechanical environment (i.e., the distribution of compressive and shear moduli) plays an important role in the cell activities within scaffolds, such as cell proliferation and differentiation [67, 68].” (Page 16 Lines 21 -23)

-Figure 3(c): What is special about the relation between compressive modulus and shear modulus? In what capacity is this important for the scaffold design?

Replies: The relation between the compressive modulus and shear modulus can help understand the anisotropic behavior of the scaffold, which can be characterized using the Zener anisotropy factor (Lu et al., 2019, JMBBM). The scaffold anisotropic behavior can help the scaffold design under the in vivo physiological loading scenario (i.e., the complex loading condition). We have added the clarification in the manuscript as below:

“It should be noted that when the scaffold is implanted into the long bone (e.g., femur), the scaffold is under the combined loading of axial compression and shear, due to the fact the femur is tilted approximately 7 degrees under the in vivo loading scenario [59]. The analysis on the compression and shear moduli of the scaffold could help derive the Zener anisotropy factor and understand the anisotropic mechanical behavior of the scaffold under the complex clinic loading scenario.” (Page 15 Lines 27 – 29 and Page 16 Lines 1 - 4)

-Table 1-3: The number of meaningful digits (digits after the decimals) should be consistent in each table.

Replies: Thanks for the comment. We have made them consistent in each table now. 

-Table 1: Please provide a % deviation of FE from experiments. The discrepancies between FE and experiments seem much less than the typical order in the literature. Can the authors provide their deviation with the literature?

Replies: Following the suggestion, we added the percentage deviation of the FE values from the experimental data into Table 2. The deviation of the FE from the experimental data is within 10% in the present study. A deviation of less than 10% in the scaffold elastic modulus for most samples is also found in Dallago et al.’s work (2018) (Figure 18 in Dallago et al.’s paper). The smaller deviation in the present study could be due to the reason that the quality of the produced scaffold is controlled by disposing the scaffolds whose porosity deviation is larger than 5% from the designed porosity. We have added the clarification in the manuscript as below:

“The defected scaffolds, i.e. the deviation of the porosity from the design value is larger than 5%, were disposed and five samples per scaffold porosity were selected.” (Page 9 Lines 23 - 25)

“Nevertheless, an acceptable discrepancy of within 10.0% is found between the FE predictions and the experimental results, which is in the same order as that reported in the literature [48].” (Page 12 Lines 24 -26) 

-The stress-strain curves for both experimental compression tests and FE simulation must be presented and compared.

Replies: Following the suggestion, we added a figure (Fig 4) to demonstrate a representative stress-strain curve from the experimental compression test. Regarding the FE simulation, only the scaffold linear behavior was simulated, so the stress-strain curves for the FE is just a linear line. 

-Table 2: Table caption is wrong. The table doesn't represent the relationship between mechanical properties, surface-to-volume ratio and permeability. Rather they show the relation between these parameters and porosity. Besides, the regression should be defined separately for each curve fitting.

Replies: Thanks for the suggestion. We have modified the table caption (as shown below) to avoid the confusion. Additionally, we have separated the regression coefficient of determination (R2) for each curve fitting. 

“Table 3. The regression equations for the scaffold mechanical properties (compressive and shear moduli), the surface-to-volume ratio and the permeability as a function of the scaffold porosity (denoted as ∅ ranged from 0.0 to 1.0)” 

-Table 3: Why E0 is "-" for Cube, FD-Cube and Octa?

Replies: Cube, FD-Cube and Octa are non-TPMS-based scaffolds. For these scaffolds, when the porosity is zero, theoretically the corresponding modulus is zero, while for the TPMS scaffolds, to guarantee the connectivity, the porosity cannot be zero. We reported “-” was to mean the values are very small and can be ignored. To avoid the confusion, we added the E0 values for the Cube, FD-Cube and Octa scaffolds. Please see the updated Table 4. 

Discussion:

-p. 13, l. 5-9: References 45 and 46 have not characterized the experimental/numerical permeability. Besides, the reported comparisons of computational and experimental permeability in the literature are highly significant with a factor of ~0.1 (see DOI: 10.1016/j.jbiomech.2012.01.019 10.1016/j.matdes.2017.03.006)

Replies: There was a mistake when organizing the reference numbers. We have now modified the references 45 and 46 to the ones recommended (i.e., 10.1016/j.jbiomech.2012.01.019; 10.1016/j.matdes.2017.03.006). Additionally, we have modified the sentences (see the updated below) to accurately convey the message:

“However, previous studies have found a highly linear correlation between the permeability derived from CFD analysis and the experimental results (with a factor of approximately 0.27), concluding that the CFD analysis is a reliable tool for estimating the scaffold permeability [50, 51].” (Page 13 Lines 2 - 5)

-p. 13, l. 9-11: The largest order of permeability in the cited papers (Ref. 47, 48), as well as the author's results (Fig. 4(b)), starts from 10^-9 (not 10^-8)

Replies: There was confusion here. The range of permeability is from 1.0 × 10-10 m2 to 1.0 × 10-8 m2 in the present manuscript and this is what we meant to compare. We have modified the sentence as below:

“This is confirmed by the fact that the range of permeability (1.0 × 10-10 m2 to 1.0 × 10-8 m2) predicted in the present study agrees with the experimental data using the flow chambers [52, 53]” (Page 13 Lines 5-7)

-p. 15, l. 29: do we have "gas diffusion" in a scaffold?

Replies: What we meant is the “oxygen diffusion”. We have changed the “gas diffusion” to “oxygen diffusion” in the manuscript:

“…and a permeable scaffold allows efficient nutrient and oxygen diffusion and waste emission through its channels [60, 61]” (Page 16 Line 8)

-p. 16, l. 15: Given the authors are mentioning the stress-shielding effect, please compare the results obtained in this study with the natural properties of bone to elucidate how far are we from mimicking the properties of native bone.

Replies: The literature data (Wirtz et al., JB, 2000) showed that the range of the Young’s modulus of femoral cortical bone is from approximately 5.00 GPa to 20.00 GPa, while the range of the Young’s modulus of femoral trabecular bone is from approximately 0.15 GPa to 1.65 GPa. In the present manuscript, the modulus of the scaffold can be tuned from approximately 5.50 GPa to 33.00 GPa. Therefore, the designed scaffold is a good candidate to replace the cortical bone to avoid the stress-shielding effect. We have added this point in the discussion part as below:

“It should be noted that the Young’s modulus of the scaffold designed in the present study can be tuned from approximately 5.5 GPa to 33.0 GPa, which make them a good candidate for mimicking the mechanical properties of cortical bone (ranged from approximately 5.0 GPa to 20.0 GPa) [57]. However, for mimicking the mechanical properties of trabecular bone (ranged from approximately 0.15 GPa – 1.65 GPa) [57], the Ti-6Al-4V scaffold is too stiff, scaffolds made from other materials such as polymer should be used.” (Page 14 Lines 10 - 17)

-p. 16, l. 21: "it should be noted that, because the TPMS sheet solids have the same microstructure topology as their network solid counterparts, the features of the mechanical and permeability properties of the TPMS sheet solids should be similar to their network solid counterparts. However, this needs to be confirmed in future studies": It is very unclear what the authors are trying to convey.

Replies: Sorry for the confusion. Another reviewer pointed out the TPMS network solid and sheet solid have very dissimilar properties, and so we have rephrased the sentences as below:

“Recent studies [72] showed that the TPMS network solid and sheet solid have very dissimilar properties. Therefore, the investigation on the TPMS sheet solids still needs to be performed in the future.” (Page 17 Lines 3 -6)

-Fig. 4(c): can you please discuss what is the reason for the huge difference between the permeability from CFD and Kozeny-Carman model and comment on which one is a more reliable model to take into account when designing scaffolds for permeability?

Replies: The Darcy’s law is based on the CFD analysis and assumes the laminar flow, while the Kozeny-Carman’s relation is an empirical relation. The reason for the big difference between the permeability from the Darcy’s law and the Kozeny-Carman model could be that the coefficient in the Kozeny-Carman’s model is not calibrated using the data from the present study. However, previous numerical and experimental studies (Dias et al., 2012; Montazerian et al., 2017) showed that the permeability from CFD only highly correlated with the experimental data with a factor of approximately 0.27. Therefore, in the authors’ opinion, the permeability from both the CFD and the Kozeny-Carman’s model needs to be calibrated to a specific application when designing scaffolds. We have added the following sentences in the manuscript:

“It also should be noted that the Darcy’s law is based on the CFD analysis where the laminar flow is assumed, while the Kozeny-Carman’s relation is an empirical one. Because no permeability test is performed in the present study, no calibration can be done for the numerically calculated permeability, which could be reason the permeability from the two methods significantly differ in the high porosity region.” (Page 16 Lines 9 - 14)

-In practice, the scaffold is placed and fit in the holes drilled in bone by the surgeon. Imagine as if a cylinder is mechanically loaded in transverse direction. Given the cubic models studied here, can you comment on how can these results can be translated and applied to the case of complex loading configurations that are applied under the physiological conditions?

Replies: Thanks for the very good point. It is well accepted that the properties of the scaffold should be close to that of the replaced bone tissues. The data on the effective compressive and shear moduli, and the permeability could help select the most appropriate scaffold topology and can also help tune the scaffold microstructure to match the property of the replaced native human tissues. On the other hand, one of our previous publications (Lu et al., JMBBM, 2019) showed that the scaffolds investigated in the present study possess cubic symmetry and have 3 independent constants. Therefore, the mechanical elastic stiffness matrix of the scaffold can be approximately estimated from the compressive and shear moduli and the elastic stiffness matrix is used to describe the mechanical behavior of scaffold in the case of complex loading configurations. We have the following sentences in the manuscript: 

“It should be noted that when the scaffold is implanted into the long bone (e.g., femur), the scaffold is under the combined loading of axial compression and shear, due to the fact the femur is tilted approximately 7 degrees under the in vivo loading scenario [59]. The analysis on the compression and shear moduli of the scaffold could help derive the Zener anisotropy factor and understand the anisotropic mechanical behavior of the scaffold under the complex clinic loading scenario.” (Page 15 Lines 27-29 and Page 16 Lines 1 - 4)

-Does the simulation parameters such as viscosity, flow rate, scaffold length, and cross-section affect computational permeability? If so, why the permeability can be a suitable parameter for correlating pore shape to biological behavior while it is cross-sensitive to the abovementioned factors?

Replies: According to the Darcy’s law, the permeability is positively linearly correlated with the fluid flow rate and the scaffold length, and negatively linearly correlated with the scaffold cross-sectional area. In the present study, the viscosity, the flow rate, the scaffold length and the cross-section area were made the same in the comparison among different scaffolds, and the investigation only at one point was made. Because the pressure drop in the scaffold depends on the scaffold microstructure and dimensions, the scaffold permeability may be disproportionally influenced by the above mentioned factors among different scaffolds. Therefore, the scaffold permeability should be investigated at multiply points to generalize the conclusions (e.g., the cubic structure has the highest permeability) made in the present paper. We have added this as a shortcoming in the present manuscript. 

“Last but not the least, the influence of scaffold microstructure on the permeability is only investigated with one set of parameters (flow rate, viscosity, etc.). The scaffold microstructure may influence the pressure drop differently, and consequently the permeability calculated from the Darcy’s law may change differently when the flow rate, the scaffold length and cross-section are changed. Therefore, in the future, the correlation between the scaffold microstructure and permeability should be investigated with more sets of data.” (Page 17 Lines 6 - 12)

-The deformation mechanism is discussed in the context of mechanical properties. Please discuss what is the implication of deformation mechanisms on biological performance?

Replies: Thanks for the suggestion. The present study revealed that the bending deformation dominated structures tend to have higher effective shear moduli and the stretching deformation dominated structures tend to have higher effective compressive moduli. There is no trend that the scaffold biological performance (permeability) is correlated with the scaffold deformation mechanism. The permeability is determined by whether the microstructural arrangement facilitates the fast flow of fluid or not. We have added this interesting point in the manuscript: 

“Regarding whether the scaffold permeability correlates with its deformation mechanism (i.e., whether the stretching deformation dominated structures have a higher permeability than the bending deformation dominated structures, or vice versa), no trend is found in the present study.” (Page 11 Line 29 and Page 12 Lines 1 - 3)

Accordingly, we have updated the references in the manuscript. All the changes in the manuscript were highlighted in yellow to help the reviewers track the changes. We hope we have clearly answered all the reviewers’ concerns. If there is still something unclear, please feel free to contact me by email. 

Sincerely,

Yongtao

Yongtao Lu, Ph.D

Department of Engineering Mechanics

Dalian University of Technology, Dalian

No.2 Linggong Road,

116024, Dalian, China

Email: yongtaolu@dlut.edu.cn; yongtaolu@hotmail.com

---

## [Decision Letter · Decision Letter 1]

11 Aug 2020

PONE-D-20-13348R1

Relationship between the morphological, mechanical and permeability properties of porous bone scaffolds and the underlying microstructure

PLOS ONE

Dear Dr. Lu,

Thank you for submitting your manuscript to PLOS ONE. After careful consideration, we feel that it has merit but does not fully meet PLOS ONE’s publication criteria as it currently stands. Therefore, we invite you to submit a revised version of the manuscript that addresses the points raised during the review process.

We look forward to receiving your revised manuscript.

Kind regards,

Yanyu Chen

Academic Editor

PLOS ONE

Reviewers' comments:

Reviewer's Responses to Questions

**Comments to the Author**

1. If the authors have adequately addressed your comments raised in a previous round of review and you feel that this manuscript is now acceptable for publication, you may indicate that here to bypass the “Comments to the Author” section, enter your conflict of interest statement in the “Confidential to Editor” section, and submit your "Accept" recommendation.

Reviewer #1: All comments have been addressed

Reviewer #2: All comments have been addressed

Reviewer #3: (No Response)

Reviewer #4: All comments have been addressed

2. Is the manuscript technically sound, and do the data support the conclusions?

Reviewer #1: Partly

Reviewer #2: Yes

Reviewer #3: Partly

Reviewer #4: Yes

3. Has the statistical analysis been performed appropriately and rigorously? 

Reviewer #1: N/A

Reviewer #2: Yes

Reviewer #3: No

Reviewer #4: I Don't Know

4. Have the authors made all data underlying the findings in their manuscript fully available?

Reviewer #1: Yes

Reviewer #2: Yes

Reviewer #3: Yes

Reviewer #4: Yes

5. Is the manuscript presented in an intelligible fashion and written in standard English?

Reviewer #1: Yes

Reviewer #2: Yes

Reviewer #3: Yes

Reviewer #4: Yes

6. Review Comments to the Author

Reviewer #1: The readability of the manuscript has been enhanced. Reviewers tried to address all the comments. I recommend the work for publication.

Reviewer #2: The authors have addressed all the issues in the revised manuscript. I think it can be accepted in the present form.

Reviewer #3: It is clear that the authors have made substantial changes to their manuscript based on the questions and comments of several reviewers. I stand by my original comment that the paper presents results which are both novel and useful, especially for the designer of scaffold structures for a range of mechanical and fluid-flow applications.

But the authors have not addressed my chief criticism from the first review; the issue of the quadratic fits for compressive modulus and shear modulus. I originally stated that these fits are not generally used; the Gibson-Ashby power laws are used instead because they have a basis in the mechanical deformation of cellular struts. I asked the authors to justify their use of quadratic fits, which they have not done. By 'justify' I mean demonstrate that the fitting function is valid for this data type by considering the underlying phenomena. The authors should either provide a valid justification for all choices of fitting functions in the manuscript, or remove those which cannot be justified.

Regarding figure 6(a), why is it that, for each scaffold type, the surface-to-volume ratio barely changes over the examined range of porosity? Surely the S/V should be far greater when the porosity takes a large value; i.e. when their is a lot of surface but very little volume. Can the authors explain the 'hump shaped' curves of 6(a)?

As a final note, the authors should consider re-writing the conclusion section, as it is not appropriate to simply have the results repeated there as bullet points. Rather, the overarching findings of the paper should be clearly summarised and placed into context.

I will be happy to review the manuscript again if these changes are made.

Reviewer #4: The majority of the concerns about the paper has been addressed. The paper can be published in PLOS ONE.

7. PLOS authors have the option to publish the peer review history of their article (what does this mean?). If published, this will include your full peer review and any attached files.

Reviewer #1: No

Reviewer #2: No

Reviewer #3: No

Reviewer #4: No

---

## [Author Response · Author response to Decision Letter 1]

14 Aug 2020

Thanks to the reviewers for their valuable suggestions. They have contributed to improve the quality of the paper. We hope the responses we provide below will answer their concerns and shed light on the unclear parts of the study.

Reviewer #3: 

It is clear that the authors have made substantial changes to their manuscript based on the questions and comments of several reviewers. I stand by my original comment that the paper presents results which are both novel and useful, especially for the designer of scaffold structures for a range of mechanical and fluid-flow applications.

But the authors have not addressed my chief criticism from the first review; the issue of the quadratic fits for compressive modulus and shear modulus. I originally stated that these fits are not generally used; the Gibson-Ashby power laws are used instead because they have a basis in the mechanical deformation of cellular struts. I asked the authors to justify their use of quadratic fits, which they have not done. By 'justify' I mean demonstrate that the fitting function is valid for this data type by considering the underlying phenomena. The authors should either provide a valid justification for all choices of fitting functions in the manuscript, or remove those which cannot be justified.

Replies: Sorry, we did not address the issue properly. Following the suggestion, we have removed the quadratic fitting between the mechanical properties and the porosity and instead focus on the use of Gibson-Ashby power law to explain the underlying mechanism. Please see the updated Table 4 and the relevant update in the manuscript. 

“Regarding the mechanical property, to reflect the underlying physical phenomena, the relationship between the relative elastic compressive modulus and the scaffold volume fraction were described using the exponential function proposed by Gibson and Ashby…”(Page 8 Lines 23 - 26)

“Regarding the scaffold permeability and the surface-to-volume ratio, the statistical regression equations (quadratic or other forms) and the coefficient of determinations (R2) were computed for the relationships between them and the scaffold porosity. The reasons for deriving these statistical regression equations are to enable the interpolation of the data points to the full scaffold porosity range and to facilitate the scaffold design by using these relations.” (Page 9 Lines 7-12)

Regarding figure 6(a), why is it that, for each scaffold type, the surface-to-volume ratio barely changes over the examined range of porosity? Surely the S/V should be far greater when the porosity takes a large value; i.e. when there is a lot of surface but very little volume. Can the authors explain the 'hump shaped' curves of 6(a)?

Replies: Thanks for the suggestion. We now added the data on the relationship between the surface-to-volume ratio and the porosity in the full range of scaffold porosity (0.0 to 1.0) (Fig. 6a). From the added figure, it is clearly shown that the surface-to-volume ratio barely changes with the porosity is not true. The reason we presented Fig. 6b is that some scaffolds will lose the connectivity when the porosity gets too big. Therefore, we chose the range (i.e., 0.3 – 0.7) in which all the scaffolds are valid and made the comparison in these meaningful range. We believe the reason for the ‘hump shaped’ curves is that: the inner surface area was used to calculate the surface-to-volume ratio (when the porosity gets lower, there are more overlapped surfaces; when the porosity gets higher, the inner surfaces get fewer), and so the surface-to-volume ratio is not a monotonic function of the scaffold porosity. Please see the updated Figures 6 and 7, and the clarification in the manuscript as below:

“For all the scaffolds except the Octa-based one, the surface-to-volume ratio is not a monotonic function of the porosity and the surface-to-volume ratios are the highest when the porosity is 0.5, and start to decrease when the porosity is away from 0.5, the reason for which could be that the overlapped inner surfaces increase when the porosities get lower and there are fewer inner surfaces with the increase of the scaffold porosity.” (Page 11 Lines 15-20)

“The interpolated values using the fitted quadratic relationships and the comparison of the surface-to-volume ratio in the porosity range from 0.3 to 0.7 are presented in Fig 6b.” (Page 11 Lines 24-26) 

As a final note, the authors should consider re-writing the conclusion section, as it is not appropriate to simply have the results repeated there as bullet points. Rather, the overarching findings of the paper should be clearly summarized and placed into context. 

Replies: Thanks for the suggestion. We have re-writing the conclusion section as below: 

“The bending dominated scaffolds (e.g., Diamond, Gyroid, Schwarz P, Fischer-Koch S and R-RD) tend to have a higher effective shear modulus. The stretching dominated scaffolds (e.g. Schwarz P, Cube, FD-Cube and Octa) tend to have a higher effective compressive modulus. 

The relative shear modulus of the scaffold changes faster than the relative compressive modulus, i.e., when the same amount of change in the scaffold porosity is made, the corresponding change in the relative shear modulus is larger than that in the relative compressive modulus. 

The permeability of the scaffold depends on the arrangement of the underlying microstructure, e.g., the structures with the simple and straight pores (e.g., Cube) have a higher permeability than the structures with the complex pores (e.g., Fischer-Koch S). ”

All the changes in the manuscript were highlighted in yellow to help the reviewers track the changes. We hope we have clearly answered all the reviewers’ concerns. If there is still something unclear, please feel free to contact me by email.

---

## [Editor Report · Decision Letter 2]

18 Aug 2020

Relationship between the morphological, mechanical and permeability properties of porous bone scaffolds and the underlying microstructure

PONE-D-20-13348R2

Dear Dr. Lu,

We’re pleased to inform you that your manuscript has been judged scientifically suitable for publication and will be formally accepted for publication once it meets all outstanding technical requirements.

Kind regards,

Yanyu Chen

Academic Editor

PLOS ONE

---

## [Editor Report · Acceptance letter]

19 Aug 2020

PONE-D-20-13348R2 

Relationship between the morphological, mechanical and permeability properties of porous bone scaffolds and the underlying microstructure 

Dear Dr. Lu:

I'm pleased to inform you that your manuscript has been deemed suitable for publication in PLOS ONE. Congratulations! Your manuscript is now with our production department. 

Kind regards, 

on behalf of

Dr. Yanyu Chen 

Academic Editor

PLOS ONE